# Fecal microbiota transplantation in HIV: A pilot placebo-controlled study

Sergio Serrano-Villar [1✉], Alba Talavera-Rodríguez[2], María José Gosalbes[3,4], Nadia Madrid[1], José A. Pérez-Molina[1], Ryan J. Elliott [5], Beatriz Navia[6], Val F. Lanza[2], Alejandro Vallejo[1], Majdi Osman[5], Fernando Dronda[1], Shrish Budree[5], Javier Zamora[7], Carolina Gutiérrez[1], Mónica Manzano [6], María Jesús Vivancos[1], Raquel Ron[1], Javier Martínez-Sanz[1], Sabina Herrera[1], Uxua Ansa[1], Andrés Moya [3,8] & Santiago Moreno[1]

Changes in the microbiota have been linked to persistent inflammation during treated HIV infection. In this pilot double-blind study, we study 30 HIV-infected subjects on antiretroviral therapy (ART) with a CD4/CD8 ratio < 1 randomized to either weekly fecal microbiota capsules or placebo for 8 weeks. Stool donors were rationally selected based on their microbiota signatures. We report that fecal microbiota transplantation (FMT) is safe, not related to severe adverse events, and attenuates HIV-associated dysbiosis. FMT elicits changes in gut microbiota structure, including significant increases in alpha diversity, and a mild and transient engraftment of donor's microbiota during the treatment period. The greater engraftment seems to be achieved by recent antibiotic use before FMT. The Lachnospiraceae and Ruminococcaceae families, which are typically depleted in people with HIV, are the taxa more robustly engrafted across time-points. In exploratory analyses, we describe a significant amelioration in the FMT group in intestinal fatty acid-binding protein (IFABP), a biomarker of intestinal damage that independently predicts mortality. Gut microbiota manipulation using a non-invasive and safe strategy of FMT delivery is feasible and deserves further investigation. Trial number: NCT03008941.

---

[1] Department of Infectious Diseases, Hospital Universitario Ramon y Cajal, and IRYCIS, Madrid, Spain. [2] Bioinformatics Unit, Hospital Universitario Ramon y Cajal, IRYCIS, Madrid, Spain. [3] Area of Genomics and Health, FISABIO-Salud Pública, Valencia, Spain. [4] Centro de Investigación Biomédica en Red Epidemiología y Salud Pública (CIBERESP), Madrid, Spain. [5] OpenBiome, Cambridge, MA, USA. [6] Department of Nutrition, Facultad de Farmacia, Universidad Complutense, Madrid, Spain. [7] Barts and the London School for Medicine and Dentistry. Queen Mary University of London, London, UK. [8] Institute for Integrative Systems Biology (I2SysBio), The University of Valencia and The Spanish National Research Council (CSIC)-UVEG), Valencia, Spain. ✉email: sergio.serrano@salud.madrid.org

HIV is a chronic inflammatory disease, in which chronic immune dysfunction, that seems to be influenced by the microbiome[1–4], leads to persistent inflammation and contributes to the excess risk of mortality[5]. For many years, mitigating the multifaceted HIV-associated gut-associated lymphoid tissue defects has been pursued in an attempt to reduce the long-term consequences of chronic inflammation through different interventions[6,7].

The mucosal immune system is devastated after acute HIV infection[5]. Today, it is widely accepted that the gut microbial communities are not passive when confronted with the HIV infection, but rather they play an active role in disease progression, including susceptibility to HIV acquisition[8], pre-exposure prophylaxis efficacy[9], chronic inflammation[1–3], and HIV-vaccine efficacy[10]. Hence, modification of the gut microbiota is being pursued as a strategy to improve clinical outcomes in people living with HIV (PWH).

Multiple studies have assessed dietary supplementation in PWH with various nutritional products, such as prebiotics and probiotics, among others[11,12]. However, the two large randomized controlled trials assessing the effects of probiotics or synbiotics -a combination of prebiotics and probiotics- on PWH on antiretroviral therapy (ART) have failed to detect differences in any of the outcomes assessed[12,13]. At the gut microbiota compositional level, we previously found that even after 48 weeks of daily synbiotic supplementation, no consistent changes in gut microbiota structure nor differences in systemic markers of inflammation or immune activation were observed compared to placebo[12]. So far, in PWH the gut microbiota has shown a high resilience to interventions and remains an elusive therapeutic target. It is unknown whether fecal microbiota transplants (FMT) delivered orally can affect the gut microbiota, as a method to attenuate chronic inflammation in PWH.

In this work, we hypothesized that repeated FMT using capsulized stools will be more efficacious than previous nutritional interventions aimed at shaping the microbiota in PWH and improve systemic markers of inflammation. Because such an intervention has not been explored before, we conducted a controlled, double-blind, placebo-controlled pilot study in which PWH on stable ART received eight courses of oral FMT or a placebo and were subsequently followed for 48 weeks. The principal outcome was safety. Secondary outcomes were exploratory and included changes in CD4 + T cell counts, CD8 + T cell counts, CD4/CD8 ratio, inflammatory markers, T cell activation, and markers of enterocyte barrier function through week 48. Because a hallmark of the microbiome abnormalities associated with HIV infection is a depletion of butyrate-producing bacteria (Lachnospiraceae and Ruminococcaceae families)[14], we specifically searched for donors with enrichment for butyrate in their feces.

## Results

**General characteristics of the study population and safety data.** Between January and May 2017, 47 participants were screened to participate in the study; 17 were not eligible and the remaining 30 were randomized into treatment or placebo groups. A total of 29 subjects completed the 48-week follow-up (Supplementary Figure 1). Their main characteristics are summarized in Table 1, and the individual metadata are provided in Supplementary Table 1. The study participants were representative of a population of middle-aged men who have sex with men with well-controlled HIV infection. Six subjects had received antibiotic treatment in the 14 weeks before the intervention, of whom three were in the FMT arm (Supplementary Table 2). No significant between-groups differences were observed in food consumption and energy and nutrient intake were observed (Supplementary Table 3).

No serious adverse events attributable to the intervention were reported, and all participants maintained virologic suppression throughout the study except subject R18, who discontinued ART between weeks 9 and 30, presenting with CD4 counts 230/uL, HIV RNA 5.2 log copies/mL and oral candidiasis at week 12, grade 2 diarrhea due to cryptosporidiosis at week 24 and latent syphilis at week 24. In the FMT arm, five individuals reported mild abdominal distension, flatulence, and diarrhea starting some hours after capsules intake until 48–72 h after. By the end of the study, two patients reported improvement of chronic constipation from the baseline. These symptoms were not appreciated in the placebo arm, in which only one subject reported increased flatulence after the first course of capsules.

**Changes in microbial alpha diversity.** Alpha diversity is used to measure the richness and evenness of bacterial taxa within a community. We found that, compared to subjects in the placebo arm, bacterial richness increased incrementally in the FMT arm from 250 OTUs at week 0 to 287 at week 6, while in the placebo arm remained stable from 252 OTUs at week 0 to 254 at week 6 (differences between treatment arms from baseline through week 7, $p = 0.014$). While this effect was attenuated during the follow-up after the FMT interventions, the intervention appeared to elicit long-lasting effects at this structural level (changes from baseline through week 6, $p = 0.011$; from week 7 to week 48, $p = 0.039$). Similar results were observed for the Chao1 estimator (which estimates the richness considering the rare taxa) and Shannon index (which considers both the number of different taxa and their abundance) (Fig. 1). The visual inspection of these plots segregated by donors indicated that the clearest increases in alpha diversity was found among participants receiving FMT from donor A (Shannon index) (Supplementary Figure 2A). Within the FMT group, we did not identify a different pattern between the four subjects with recent antibiotic exposure and the remaining ten patients (Supplementary Figure 2B).

**Engraftment of donor's microbiota on study participants.** At baseline, Principal Coordinates Analysis (PCoA) representing beta diversity distances showed no differences between participants in the FMT and placebo groups (Supplementary Figure 3, Adonist test, $p = 0.442$). During the follow-up, microbiome shifts were more pronounced in the FMT group than in the placebo group, as shown in PCoA representing donor microbiota profiles and HIV-infected recipient community dynamics with FMT or with placebo (Supplementary Video 1) and by comparison of within-group beta diversity dispersions (Supplementary Figure 4). To understand the extent of engraftment of donor's microbiota on the recipients, we first analyzed the Unifrac distances from recipients to donors at each time point, as an indicator of changes in the number of taxa shared between donors and patients (Fig. 2A). For unweighted Unifrac, which considers the presence/absence of taxa, the peak effect occurred at week 7 without evidence of a plateau (fold change, −4% in the FMT arm vs +1% in the placebo arm) and returned to baseline values after week 8 (FMT vs. placebo trajectories until week 8, $p = 0.549$; beyond week 8, $p = 0.829$). In contrast, for weighted Unifrac, which also considers the abundance of the taxa differentially abundant and their phylogenetic relatedness, the effect was incremental after each FMT, peaked at week 8 (fold change, −7% in the FMT arm vs +7% in the placebo arm; FMT vs. placebo trajectories until week 8, $p = 0.029$; beyond week 8, $p = 0.995$). These findings are indicative of a small and transient but detectable effect at the beta diversity level, and suggests that greater engraftment could have

**Table 1 General characteristics of the study participants.**

| | FMT | Placebo | Total |
|---|---|---|---|
| | n = 14 | n = 15 | n = 29 |
| Age, mean (SD), years | 48 (10) | 46 (12) | 47 (11) |
| Male, n (%) | 12 (86) | 13 (87) | 26 (90) |
| Ethnicity | | | |
| Caucasic | 13 (93) | 8 (53) | 21 (72) |
| Latinamerican | 1 (7) | 5 (33) | 6 (21) |
| African Sub-Saharian | 0 (0) | 2 (13) | 2 (7) |
| Body mass index, median (P25-P75), kg/m2 | 24 (20–26) | 25 (23–27) | 24 (22–26) |
| Years since HIV diagnosis, median (P25-P75) | 19 (7–26) | 5 (3–10) | 7 (3–19) |
| Risk factor, n (%) | | | |
| MSM | 9 (64) | 11 (73) | 20 (69) |
| HTX | 0 (0) | 3 (20) | 3 (10) |
| IDU | 5 (36) | 1 (7) | 6 (21) |
| AIDS diagnosis, n (%) | 4 (29) | 3 (20) | 7 (24) |
| HIV RNA < 20 copies/mL | 15 (100%) | 15 (100%) | |
| Nadir CD4, median (P25-P75), cells/uL | 110 (56–187) | 259 (160–300) | 180 (106–284) |
| CD4 + T cell counts, median (P25-P75), cells/uL | 703 (457–852) | 545 (454–684) | 582 (455–768) |
| CD4/CD8 ratio, median (P25-P75), cells/uL | 0.58 (0.45–0.78) | 0.73 (0.39–0.96) | 0.64 (0.43–0.86) |
| HCV positive, n (%) | 6 (43) | 3 (20) | 9 (31) |
| Use of antibiotic in past 6 months, n (%) | 3 (21) | 4 (27) | 7 (24) |
| On triple ART, n (%) | 14 (100) | 15 (100) | 29 (100) |

Abbreviations: IDU, injection drug use; MSM, men who have sex with men; HTX, heterosexual. All patients HCV positive had undetectable HCV RNA levels.

been achieved with additional FMT courses. As shown in Fig. 2B, the effect on weighted Unifrac distances was again more pronounced for donor A. While the use of antibiotics after the baseline did not introduce drastic changes in the microbiota, as shown by the individual weighted Unifrac distances from baseline to donors, donor's microbiota engraftment on participants in the FMT arm was more apparent in the 4 subjects who were exposed to antibiotics either before (n = 3) or early after the first FMT (n = 1) at both alpha (Supplementary Figure 2B) and beta (Supplementary Figure 2C) diversity levels. Figure 3 represents the individual weighted Unifrac distances from baseline to donors. Subjects R1 and R2 showed the greatest engraftment, being the most prominent for subject R2, who received a 7-day course of amoxicillin/clavulanate until the day before of the first FMT course. The use of antibiotics after the baseline did not introduce drastic changes in the microbiota.

**Changes in microbiome taxonomic composition throughout the study.** We observed heterogeneity in the microbiota profiles defined by the top 15 most prevalent genus across participants, and these compositional features were rather stable within participants during 48-week the follow-up (Fig. 4). As another approach to assess the disturbance on the recipients' microbiome elicited by repeated FMT, we used the linear discriminative analysis (LDA) effect size (LEfSe) biomarker discovery tool, a method that addresses the challenge of finding microorganisms that consistently explain the differences between two or more microbial communities[15]. Numerically, the number of taxa driving differences in microbiota composition across key timepoints (weeks 1, 3, 5, 8, and 48) drastically differed in the placebo arm (range, 0–3) compared to the FMT arm, in which six bacterial biomarkers were identified at week 1, 16 at week 3, 26 at week 5, and 16 taxa were still found to drive differences at week 48 (Supplementary Figure 5). To further understand the dynamic interactions between the donors' and recipients' microbiota across study visits, we constructed heatmaps by plotting the LDA distances from baseline to each time point for each putative biomarker identified by LEfSe in relation to the abundance in donor's microbiota (Fig. 5). In the placebo arm, we detected no

taxa consistently affected across study visits, compared to their baseline abundance. In clear contrast, we detected enrichment of several taxa in the FMT arm over time, which could even be appreciated at the last timepoints, reinforcing the observation that repeated FMT interventions caused long-lasting effects in the recipients' microbiome. Specifically, several members of the Lachnospiraceae family were significantly enriched over time in the FMT group, including *Anaerostipes spp.*, *Blautia spp.*, *Dorea spp.*, and *Fusicatenibacter spp.*, which was the genus more consistently engrafted among those enriched in the three donors. Also, different members of the Ruminococcaceae family were also consistently detected across the study visits in the FMT arm. We found that the significant increase of Ruminococcaceae and Lachnospiraceae families in the FMT arm was driven by engraftment of these taxa in the majority of individuals, rather than being artificially driven by a dramatic increase in a subset of individuals (Supplementary Figure 6).

**Effects of repeated FMT of T-cells and plasma biomarkers.** Circulating CD4 + and CD8 + T cells, the CD4/CD8 ratio, markers of immune competence (CD4 + CD8 + T-cell count, CD4/CD8 ratio), immune activation (%HLADR + CD38 on CD8 + T cells) and senescence (CD28- and PD-1+ on CD8 + T cells) (Supplementary Figure 7) and plasma markers of inflammation and bacterial translocation (Fig. 6). were assessed across time, and no trends were evident between groups. We found, however, an early significant 0.5-fold-change decrease of IFABP, already apparent at week 1 (FMT vs. placebo, p = 0.063), which reached statistical significance at week 4 (FMT vs. placebo, p = 0.040) and remained stable until week 48 (FMT vs. placebo, p = 0.013)(Fig. 6H and Supplementary Figure 8). Interestingly, this decrease was especially pronounced for patients randomized to donor A (Fig. 6J and Supplementary Figure 8).

**Discussion**

In this controlled study evaluating repeated oral FMT in PWH, we found that the intervention was safe and well-tolerated, and anecdotally, improved chronic constipation in two participants. In contrast with previous dietary interventions with prebiotics,

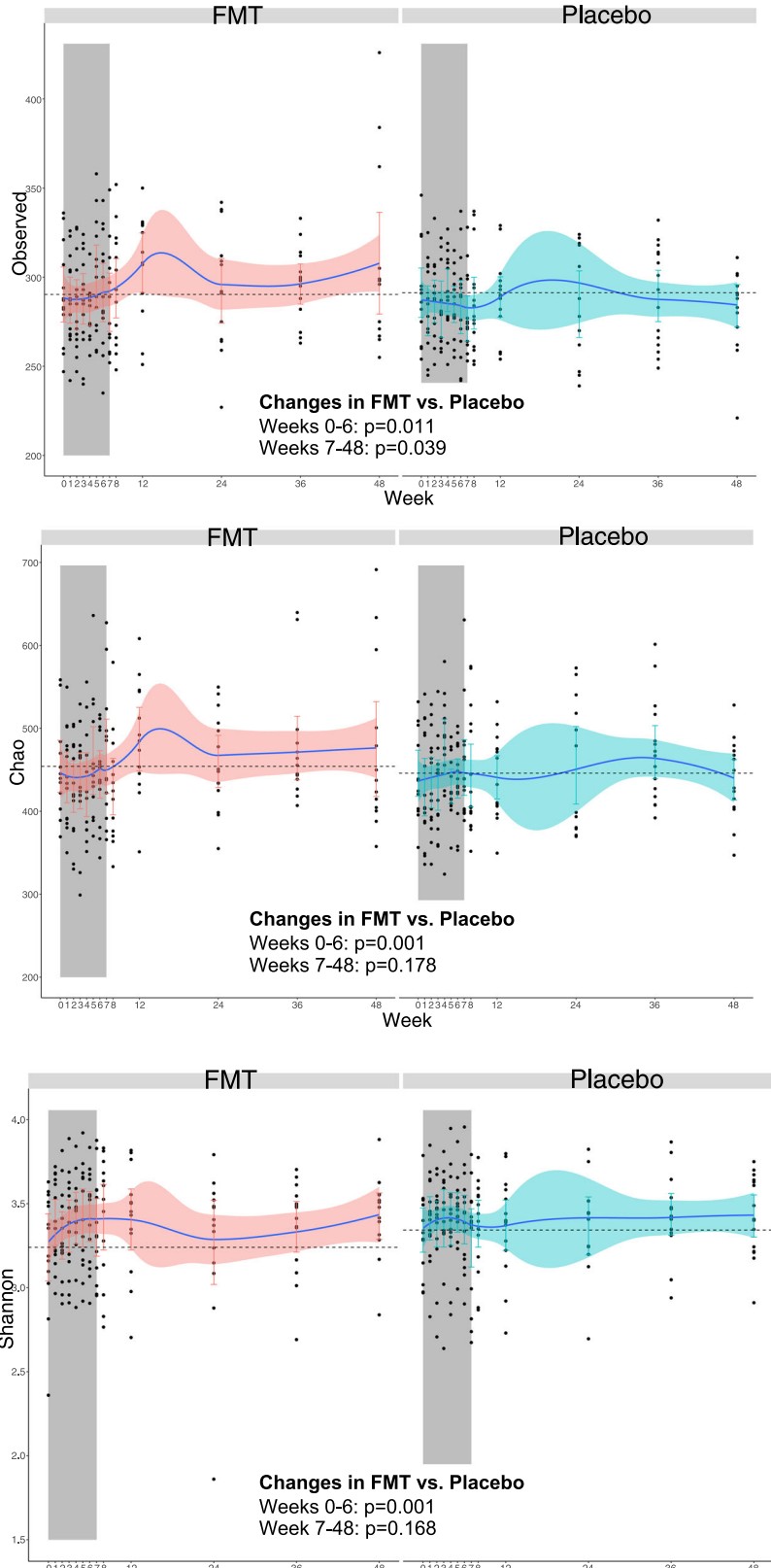

**Fig. 1 Changes in three metrics of alpha diversity at the OTUs level in each the FMT (red) and placebo (blue) groups.** Bacterial richness (number of OTU), Chao1 and Shannon indexes increased incrementally in the FMT arm. Black dots represent individual measurements. Blue lines represent the smoothed mean value. Horizontal dashed lines represent the baseline levels. Vertical bars represent the 95% confidence intervals. The gray area indicates the induction period in which study participants received FMT or placebo. Two-sided $P$ values estimated using mixed models not adjusted for multiple comparisons remained unchanged after adjustment for nadir CD4 and time since HIV diagnosis. Abbreviatures: FMT, fecal microbiota transplant. $n = 361$ biologically independent samples from 14 individuals in the FMT group and 15 individuals in the placebo group.

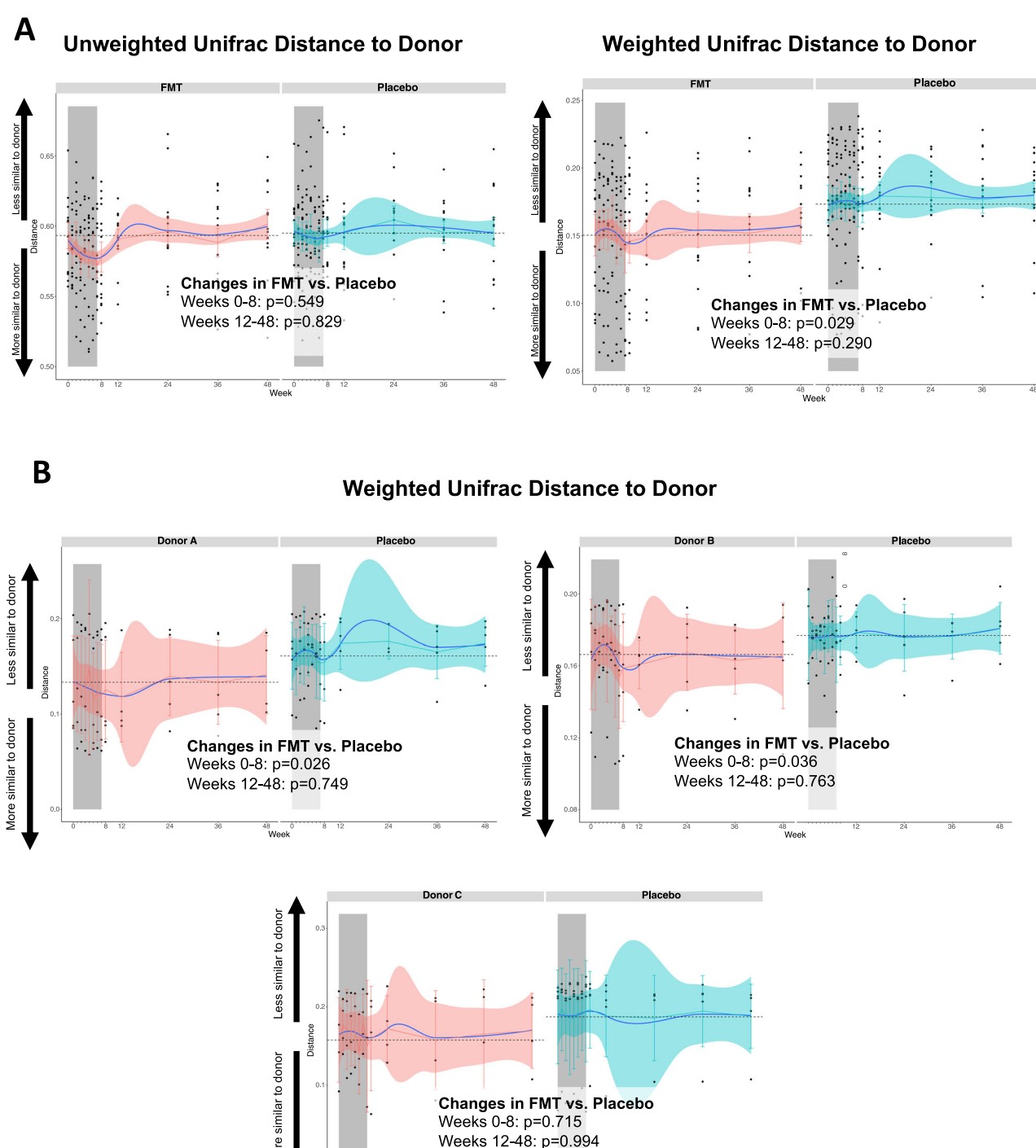

**Fig. 2 Engraftment of donor's microbiota on study participants. Unifrac distances from recipients in the FMT (red) and placebo (blue) groups to donors.** Panel **A** represents the unweighted and weighted Unifrac trajectories in the FMT and the placebo group. The peak effect occurred at week 3 for unweighted Unifrac, which considers the presence or absence of taxa, and at week 8 for weighted Unifrac, which considers their abundance. To explore donor-effects, in panel **B** we segregated the subjects in the FMT group per donor. The effect on weighted Unifrac distances were more pronounced for donor A. The Unifrac distances in the placebo group were calculated from each sample to the assigned donor, in order to understand the similarities with donors explained by randomness. Black dots represent individual measurements. Blue lines represent the smoothed mean value. Horizontal dashed lines represent the baseline levels. Vertical bars represent the 95% confidence intervals. The gray area indicates the induction period in which study participants received FMT or placebo. Two-sided *P* values estimated using mixed models not adjusted for multiple comparisons remained unchanged after adjustment for nadir CD4 and time since HIV diagnosis. Abbreviations: FMT, fecal microbiota transplant. n = 361 biologically independent samples from 14 individuals in the FMT group and 15 individuals in the placebo group.

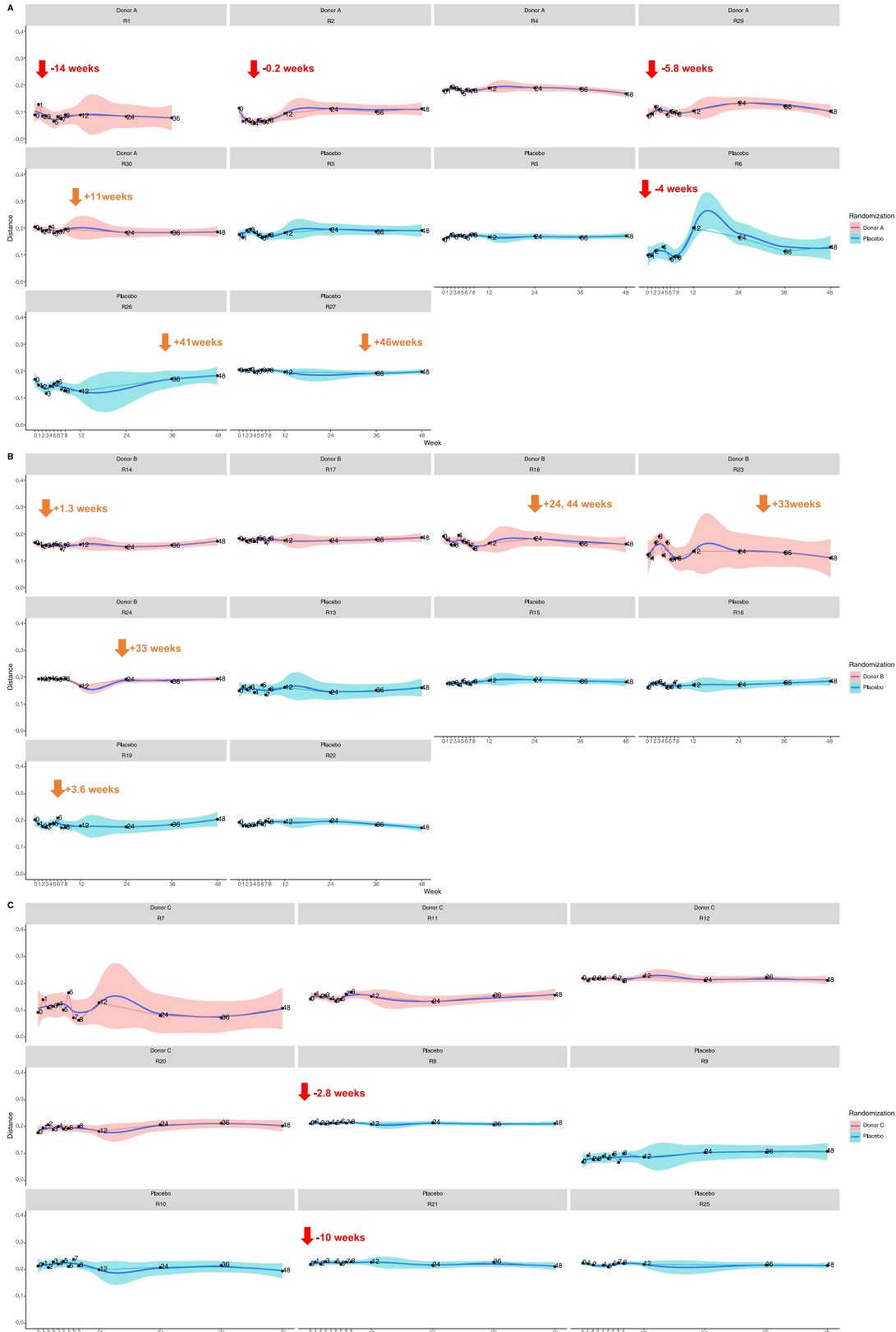

**Fig. 3 Engraftment of donor's microbiota on study participants. Individual weighted Unifrac distances from recipients to donor.** Individual weighted Unifrac distances from recipients to donors **A**, **B** and **C** (in red) have been represented together in panels **A**, **B**, and **C**, respectively. Black dots represent individual measurements. Blue lines represent the smoothed mean value. Shadow areas represent the 95% confidence intervals estimated by the smooth function. Red arrows indicate the use of antibiotics before the baseline, orange arrows indicate the use of antibiotics after the baseline. The Unifrac distances among patients in the placebo arm (in blue) were calculated using the mean values to a donor randomly assigned. Detailed information regarding the type of antibiotic, dosage, duration and indication is provided in Supplementary Table 2.

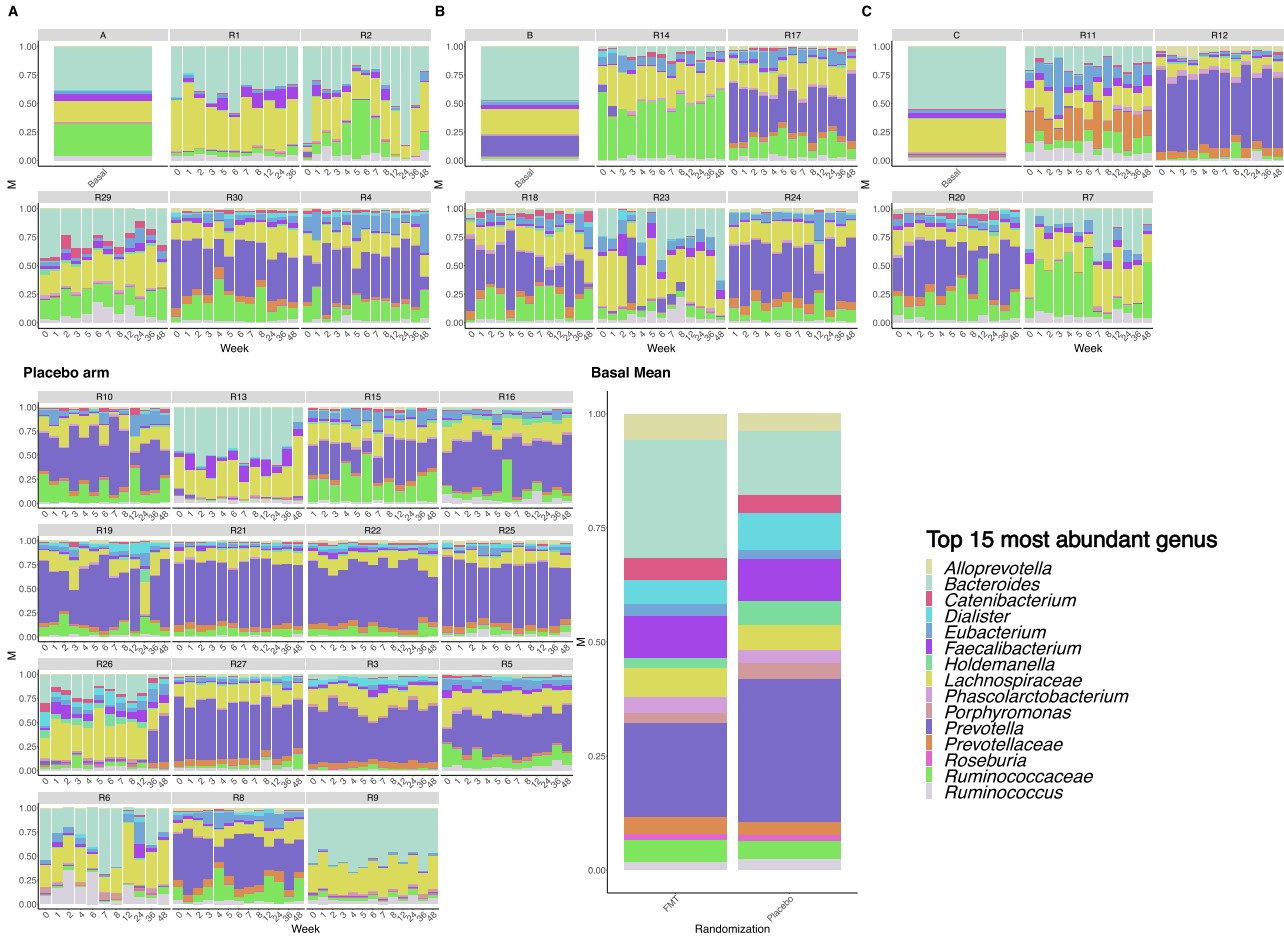

**Fig. 4 Taxonomic composition of the top 15 most abundant genus.** Each plot represents one study subject. Within each plot, bars represent independent measurements over time. Individuals grouped by donor **A**, **B** or **C**, or placebo are represented in the respective panels, in which the columns **A**, **B** and **C** denote the microbiota composition of donors **A**, **B** and **C**, respectively. The panel "basal mean" represents the mean abundance at baseline in the FMT and placebo groups. $n = 369$ biologically independent samples from 14 individuals in the FMT group, 15 individuals in the placebo group, and 3 donors.

probiotics, and synbiotics[6,11,16], we found that it is possible to induce long-lasting changes on the gut microbiota through an intervention targeting the gut microbial ecology. While the intervention did not cause any signal of immunological harm, an early decline of IFABP (a biomarker of intestinal injury which independently predicts mortality in treated PWH)[17,18] was observed in subjects receiving FMT compared to those receiving a placebo. This finding suggests that oral FMT is a candidate intervention to evaluate in studies aimed at targeting a critical root of chronic inflammation (i.e., microbial translocation), which results from the breach in the integrity of the mucosal immune system secondary to acute HIV infection[19]. Because no further improvement in any the other biomarkers representing different pathways of chronic inflammation was detected, it could happen that the microbiome is not causally linked with the other pathways of inflammation measured in this study or, alternatively, more drastic changes may be needed to affect these inflammatory outcomes.

Several findings regarding the changes in the microbiota achieved through repeated FMT deserve consideration. First, most (but not all) studies in HIV have found that HIV infection is associated with a decreased alpha diversity[20], a feature negatively correlated with immune status[21]. We detected a progressive and significant increase in the alpha diversity of the microbiota in the FMT group. The effect appeared to last until the last visit, 48 weeks after the first FMT, suggesting engraftment of

donor's microbiota and a long-lasting beneficial effect at this ecological level.

Second, the alpha diversity analysis also suggested a greater disturbance on the participant's microbiota was achieved with donor A. Because all three donors were selected based on a common microbiota signature, we sought to explore additional factors that could explain this greater effect for donor A. The three donors had microbiome profiles with a predominance of *Bacteroides*, Lachnospiracease, and *Faecalibacterium*. The most salient feature of donor A, compared to donors B and C, was a greater abundance of the Ruminococcaceae family. Also, three out of five recipients of donor A had received antibiotics in a window of 14 weeks before the first FMT, suggesting that this factor could also have influenced greater engraftment. To what extent the microbiota is resilient to FMT in chronic conditions is poorly understood, although we know that the microbiomes in stable configurations (e.g., a controlled chronic disease, such as treated HIV) show a higher resilience than those in unstable configurations (e.g., acute diseases such as *Clostridoides difficile* diarrhea)[22]. Vujkovic-Cvijin et al. previously explored the effect of a single FMT delivered by colonoscopy in 6 PWH[23]. While microbial engraftment was measurable in this study, not all subjects experienced engraftment, which was more pronounced among the individuals with lower alpha diversity at baseline, and the effect tended to disappear during the 24-week follow-up. Because our data suggested potential association between recent

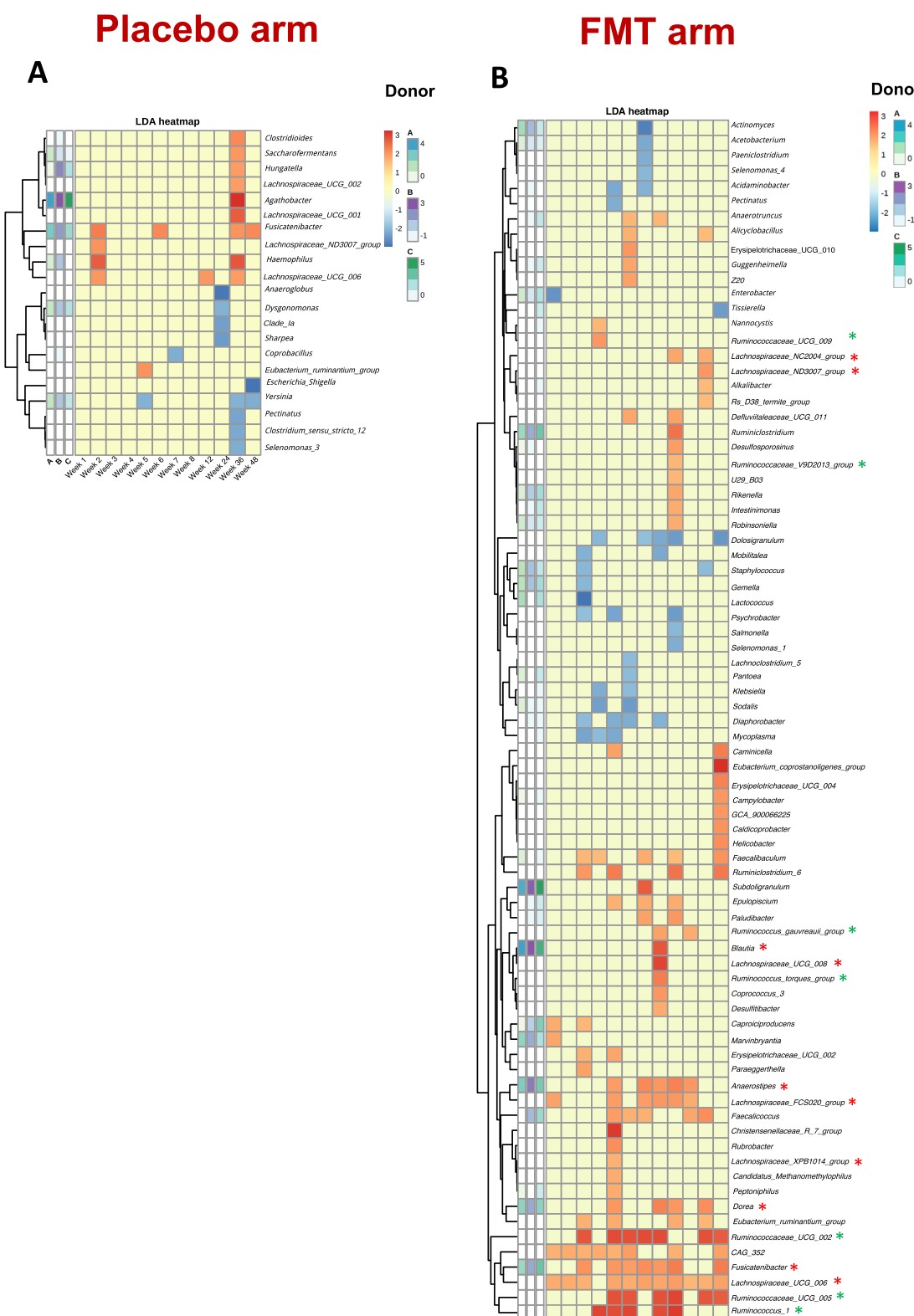

**Fig. 5 Heatmap of LDA distances at the genus level from study participants to donors at each time point in the placebo (panel A) and FMT (panel B) groups.** For the LDA distances calculations in the participants allocated to receive a placebo, the values represented were calculated as the distances from each timepoint to the assigned donor. Each column represents one timepoint. The three columns represented on the left of the pale-yellow cells represent the distance from the assigned donor to the baseline visit, and inform on the relative abundance of the represented genus on study participants with respect to donors at baseline. Green asterisks identify the genus belonging to the Ruminococcaceae family. Red asterisks identify the genus belonging to the Lachnospiraceae family. Abbreviatures: FMT, fecal microbiota transplant. $n = 369$ biologically independent samples from 14 individuals in the FMT group, 15 individuals in the placebo group, and 3 donors.

# Plasma biomarkers of inflammation and bacterial translocation

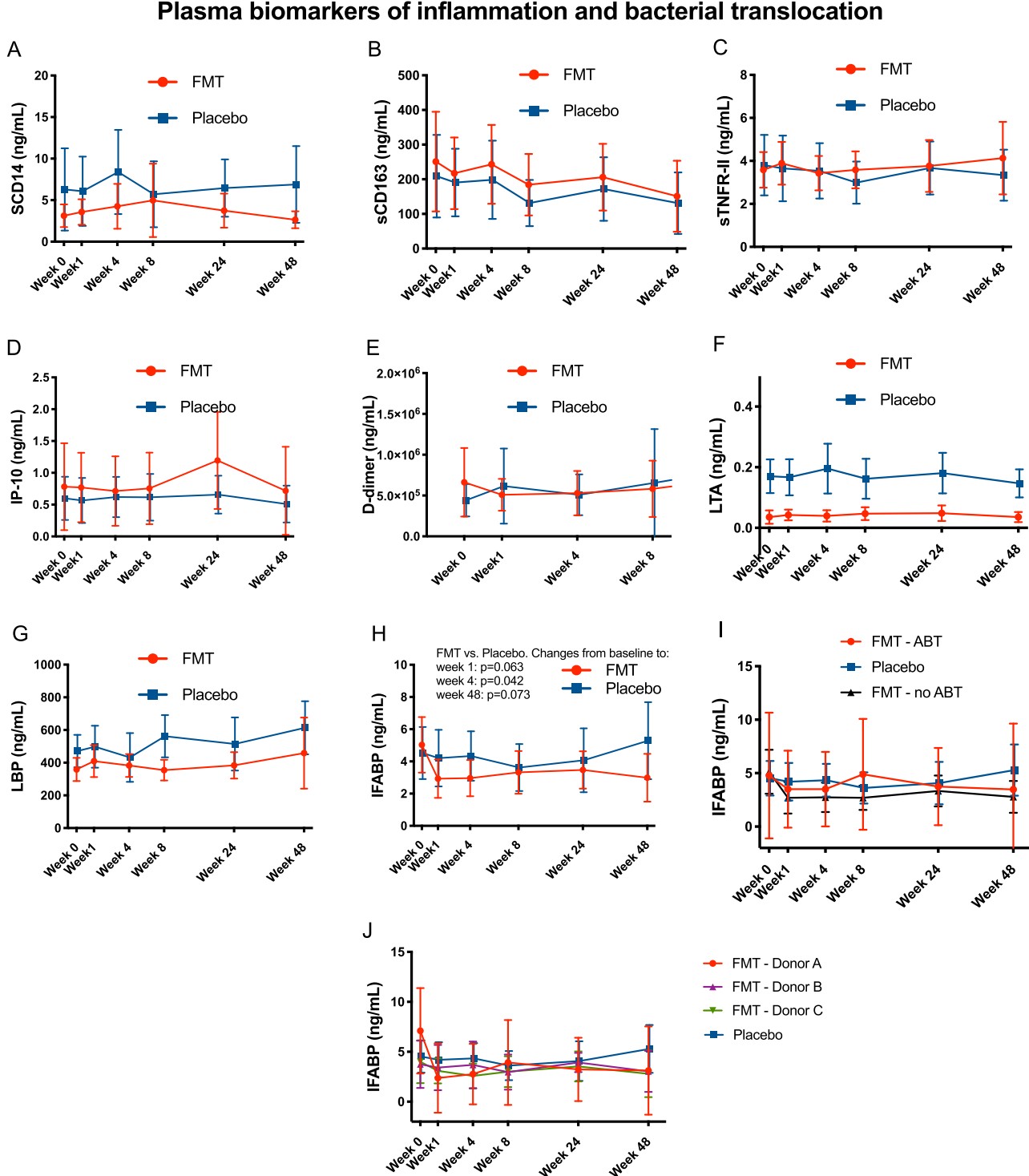

**Fig. 6 Changes in plasma biomarkers of inflammation (A: sCD14, B: sCD164, C: sTNFR-II, D: IP-10, E: D-dimer) and bacterial translocation (F: LTA, G: LBP, H, I and J: IFABP).** Lines represent mean values in the FMT (red) and placebo (blue) group. Vertical bars represent the 95% confidence intervals. Two-sided *P* values estimated using mixed models not adjusted for multiple comparisons showed no significant differences between treatment arms, unless otherwise indicated, and remained unchanged after adjustment for nadir CD4 and time since HIV diagnosis. Abbreviatures: FMT, fecal microbiota transplant. *n* = 172 biologically independent samples from 14 individuals in the FMT group and 15 individuals in the placebo group. Experiments were run in triplicate.

antibiotic exposure and greater microbial engraftment, our study further supports the notion that destabilizing the microbiota with preconditioning treatment (e.g., non-absorbable antibiotics) will be important for future studies in this field.

The third striking observation concerns the changes in beta diversity after each FMT. Unifrac distances analysis from baseline to donors indicated that incremental engraftment of donor's microbiota on recipients occurred, without a clear threshold

effect, suggesting that stronger engraftment could have been achieved if additional FMT had been given. Besides, this effect was limited to the treatment period, indicating that boosting doses may be required to maintain long-lasting effects at the beta diversity level. This observation is also relevant to the design of future studies in this direction.

The last important consideration is related to the taxonomic changes during the study. Our group previously established a link between the intestinal microbial ecology and systemic inflammatory predictors of disease progression[3,24]. The production of butyrate, a short-chain fatty acid (SCFA), by intestinal bacteria, seemed to be relevant for this interaction[7]. Across the numerous studies designed to characterize the microbial compositional features specifically associated with HIV infection, the most salient features include depletion of two families, which are the major SCFA-producers: the Ruminococcaceae and Lachnospiraceae families[14]. SCFAs are microbial-generated metabolites from dietary fiber. Beyond their role as local substrates for gut bacteria and epithelial cells, SCFAs have several immunomodulatory functions, and their effects on host physiology and several diseases continue to be revealed[25]. Butyrate is critical for maintaining enterocyte barrier integrity and mucin production[26], promotes immunotolerance to commensal bacteria[27], and induces transcription of human genes via histone deacetylase inhibition[27]. Species from the Lachnospiraceae family are considered beneficial as they produce butyrate from the digestion of dietary fiber[28] and have previously been shown to be depleted in HIV-infected subjects[1,29]. Recently, depletion of Lachnospiraceae in PWH has been shown to be associated with metabolic syndrome and visceral adipose tissue accumulation[30]. In keeping with the findings in inflammatory bowel disease[31], it has been found that *Faecalibacterium prausnitzii*, a dominant butyrate-producer in the healthy gut, is severely depleted in HIV-infected individuals, in association with an altered SCFA profile[7]. Hence, we selected donors with a particularly high abundance of the *Faecalibacterium* genus and butyrate concentrations in their stools. The finding that using FMT from butyrate-rich donors achieved a significant enrichment of the taxa containing the major butyrate-producer, supports the assertion that the intervention shaped the microbiome towards a beneficial compositional signature.

The main strengths of our study include i) the inclusion of a placebo arm that allowed to interpret to what extent the changes in the FMT arm were stochastic variations in microbial ecology rather than caused by repeated FMT, ii) the repeated measurements over time, iii) the long follow-up time frame of 48 weeks, which enabled us to understand the temporal scope of the observed changes, and iv) the selection of donors based on a microbiota profile that we considered more appropriate for this intervention. However, during the development of the study it was established that the Prevotella/Bacteroides predominance is characteristic of MSM rather than of PWH[32,33]. Still, our donor selection criteria were compatible with an anti-inflammatory microbiota profile. More importantly, we selected donors enriched in *Faecalibacterium spp.* and butyrate, robustly been associated with anti-inflammatory effects[34]. The main limitations are inherent to exploratory studies, such as a small sample size of 30 subjects. There were some differences in the baseline characteristics between the study groups, such as the time sin HIV diagnosis, the nadir CD4 count or the Prevotella/Bacteroides genus abundance. Because the rationale of the placebo group was to inform the stochastic changes in the microbiota in the absence of interventions aimed at shaping the gut microbial communities, to allow a better interpretation of the FMT effects, we do not think that this factor could have confounded the results, although this cannot be completely ruled out. We measured 8 biomarkers of inflammation, and found differences between groups only in 1

of them, which could be due to spurious associations, given the small sample sizes assessed and the lack of correction by multiple comparisons. Our study was not designed to study the effect of antibiotic preconditioning treatment, and the antibiotics received by the 4 subjects near the first FMT administrations included different antibiotics administered during a wide time frame. Hence, our findings regarding the effects of antibiotic treatment on donor's microbiota engraftment on participants must be interpreted with caution. Importantly, we did not assess the functional level of the microbiota, which we have planned to do in the near future through shotgun metagenomics and metabolomic fingerprinting.

In summary, repeated oral capsular FMT from rationally selected donors was safe in PWH on ART and introduced incremental compositional changes in the microbiota. The intervention ameliorated a marker of gut permeability that independently predicts mortality in HIV infection. Our results indicate that manipulation of the gut microbiota using a non-invasive and safe strategy of FMT delivery is feasible and encourages further research in this field. Our study supports scaling-up this intervention to larger controlled studies, possibly using antibiotic preconditioning treatment, and increasing the total FMT dose administered over time.

## Methods

**Study design, participants, setting and eligibility.** This is a randomized, double-blind, placebo-controlled pilot study (REpeated Fecal microbiota REStoration in Hiv –REFRESH–). Participants were recruited from the HIV unit of Hospital Universitario Ramón y Cajal in Madrid, Spain between January 27 and June 29, 2017. Participants were PWH on stable ART with plasma HIV RNA < 37 copies/mL during at least 48 weeks and a CD4/CD8 ratio <1, as an indicator of persistent immune defects[35]. Exclusion criteria were age <18 years, pregnancy, planned use of chemotherapy or antibiotics, neutropenia <500 cells/µL or CD4 counts <350 cells/µL, active infections or dysphagia.

The study was approved by the Ethics Committee (approval number: 165/16) and all participants signed an informed consent before the initiation of study procedures. Clinical Trials Registry Identification Number Identifier (clinicaltrials.gov): NCT03008941.

**Data collection and monitoring.** Study data were collected and managed using REDCap electronic data capture tools hosted at Fundación SEIMC-GESIDA[36,37]. REDCap (Research Electronic Data Capture) is a secure, web-based software platform designed to support data capture for research studies, providing 1) an intuitive interface for validated data capture; 2) audit trails for tracking data manipulation and export procedures; 3) automated export procedures for seamless data downloads to common statistical packages; and 4) procedures for data integration and interoperability with external sources. Data collection, study procedures and report of adverse events was monitored by the *Fundación SEIMC-GESIDA* (http://fundacionseimcgesida.org).

**Donor screening and FMT capsule preparation.** Donor screening and FMT preparation were performed during the study by OpenBiome, a nonprofit stool bank that seeks to expand safe access to FMT and catalyze research on the human microbiome[38]. OpenBiome' s screening, processing, storage and shipping controls have set the standard of care for FMT preparation production. From the moment donors donate their material, standardized, controlled processes that have been carefully evaluated by an independent panel of clinical experts are followed. OpenBiome's protocols regarding the quality and safety program, stool collection and production controls, quality assurance, and continuous donor health monitoring are available online[39]. From donors available in OpenBiome's stool, we selected three donors in the higher quartile of fecal *Bacteroides* and *Faecalibacterium* genus abundance and butyrate concentrations, and in the lower quartile of *Prevotella* abundance. This decision was made because an emerging consensus at the time of donor selection recognized that the opposite microbiota signature (i.e., enrichment for *Prevotella* genus and depletion of *Bacteroides* and *Faecalibacterium* genus) was characteristic of PWH[1,24,29,40]. The reason to select three donors with similar microbiota profiles was to be able to explore different donor effects. The placebo capsules contained glycerol, cocoa butter, and inert, non-toxic brown pigment (in place of stool). They were produced using the same protocol as active capsules.

**Intervention.** FMT was delivered by orally administered capsules versus a placebo. The dosage included induction with 10 capsules (single dose), followed by weekly maintenance FMT with five capsules for 7 weeks (see a scheme of study design in

Supplementary Figure 9). The capsule intake was supervised by our research nurse. In total, each subject received 45 capsules, implying the delivery of 30 g of stools over 8 weeks in the FMT arm. Because this was the first experience of oral FMT in PWH, we decided to explore repeated low dose FMT rather than a single FMT dose for safety reasons, being the lead in dose smaller than that typically used for the treatment of recurrent *Clostridioides difficile* infection (30 capsules). In addition, this strategy is supported by ecological principles, which dictate that gradual and persistent perturbations in the microbiome (i.e., repeated FMT) can result in long-lasting changes in the configuration of a microbial ecosystem[22].

**Study Outcomes**. After screening for eligibility, study visits were scheduled at baseline, and at weeks 1, 2, 3, 4, 5, 6, 7, 8, 12, 24, 36 and 48. At each visit, a clinical evaluation was performed, blood and fecal samples were collected, and any adverse events were registered. A dietary assessment was performed by a nutritionist. Our primary outcomes were safety and tolerability. Our secondary outcomes were changes in CD4 + and CD8 + T cells and CD4/CD8 ratio from baseline to week 48. Gastrointestinal tolerance was assessed using a questionnaire and scoring the severity of gastrointestinal symptoms on a 4-point scale and stool characteristics using the Bristol stool chart. We measured variations in the microbiota composition (alpha and beta diversity metrics) and engraftment of donor's microbiota on recipients (Unifrac distances from recipient samples to donor's microbiota).

We analyzed changes in systemic biomarkers based on their independent predictive value of mortality or their relevance in HIV immunopathogenesis[17,18]. Exploratory markers included changes in the plasma markers of inflammation (C Reactive Protein [CRP], soluble CD163 [sCD163], soluble tumor necrosis factor receptor II [sTNFr-II], monocyte/macrophage activation (soluble CD14 [sCD14], interferon-gamma induced protein 10 [IP-10]), enterocyte integrity (intestinal fatty acid binding protein [IFABP]), bacterial translocation (lipoteichoic acid [LTA], lipopolysaccharide-binding protein [LBP]), coagulation (D-dimers), T cell activation markers (% of HLADR + CD38 + T-cells), T-cell senescence markers (% of CD28- T-cells), and T-cell exhaustion markers (%PD-1+ of T-cells).

**Randomization**. The study participants were randomly assigned to active or placebo in blocks of three (one per donor) by a computer-generated randomized number system (Supplementary Figure 9). FMT and the placebo were packaged in identically appearing capsules. The bottles containing the capsules were labeled by two numeric codes. The first one corresponded to the assigned intervention. The second one corresponded to the donor (A, B and C), and was assigned consecutively to study participants at the screening visit assigned (FMT or placebo). Hence, even subjects in the placebo arm had an assigned donor, which was used to understand the similarities explained by stochasticity between subjects in the placebo arm and donors. The patient, the clinicians and nurses who attended the study participants, and investigators who handled patient specimens and performed the laboratory measurements, were blind to the assigned patient group. Only a third party had access to the randomization table and revealed the allocation to the investigators only after the statistical analyses were performed.

**Food intake and dietary assessment**. A validated three-day dietary record (from Sunday to Tuesday) was used to determine all foods and beverages consumed[41]. Briefly, participants were instructed to record all the food, beverages, and supplements consumed, including the methods for food preparation, recipes, bread, sweeteners, or snack consumption. All survey interviews were reviewed by the study dietitians (BN and MM) to assess unrealistic portion sizes, inadequate details, and typing errors, which were reassessed by a telephonic interview.

Energy and nutrients intake from the food and beverages consumed were calculated using the DIAL software version 3.0.0.12 (Alce Ingeniería, Madrid, Spain), and the data from the Spanish Food Composition Tables[42]. The values obtained were compared to those recommended[43]. In addition, the percentages of energy to total energy intake contributed by macronutrients, saturated fat, polyunsaturated fat, monounsaturated fat, omega 3 fatty acids, omega 6 fatty acids, trans-fatty acids, and sugars were calculated. The healthy dietary index[44] was also calculated, considering the specific dietary guidelines for the Spanish population[45]. Participants reported the frequency of consumption, through the *Food Frequency Questionnaire*, in the last year. The data obtained served to categorize individuals according to their averaged food consumption, and to validate the information obtained with the first dietary survey[46].

**Nucleic acid purification, amplification of the 16 S rRNA gene, sequencing, and bioinformatics analysis**
*Nucleic acid purification*. Fecal samples were stored in Omnigene Gut kits (DNA Genotek), which contain a stabilizer solution that better preserves (relative to RNAlater and Tris-EDTA) the composition of fecal microbial community structure DNA for microbiome analysis[47]. Fecal samples were aliquoted and cryopreserved at −80 °C until use.

*Amplification of the 16 S rRNA gene*. Total DNA was extracted from fecal samples in the robotic workstation MagNA Pure LC Instrument (Roche) using the MagNA Pure LC DNA isolation kit III (Bacteria, Fungi) (Roche). Total DNA was quantified with a Qubit Fluorometer (ThermoFisher). For each sample, the V3-V4 regions of

the 16 S rRNA gene were amplified and the amplicon libraries were constructed following Illumina instructions (Supplementary Table 4, Illumina, San Diego, CA, USA)[48]. The sequencing was performed using the kit V3 (2 × 300 cycles) with MiSeq sequencer (Illumina) at the FISABIO Sequencing and Bioinformatics Service, Valencia, Spain. We obtained an average of 62,939 16 S rRNA joined sequences per sample.

*Preprocessing and quality control*. All the sequences used in this analysis passed quality control, where the length and quality of the reads were filtered using the *trimmomatic* v0.33 (Paired End method, minimum length of 100, average quality of 30.)[49] Outliers were eliminated with *seqkit* v0.11.0[50]. To standardize the number of reads in the diversity analyses we used subsampling methods (seqkit, subcommand *sample*) was performed based on the minimum number of reads per sample (subcommand *stats*, 20864).

*16 S RNA gene analysis*. Amplicon data from the 16 S rRNA gene was analyzed using the taxonomic sequence classifier Kraken (v2.0.7-beta, paired-end option)[51,52], which examines the k-mers within a query sequence and uses the information within those k-mers to query a database. That database maps k-mers to the lowest common ancestor of all genomes known to contain a given k-mer. Taxonomic information on the 16 S rDNA sequences was obtained using the Silva ribosomal RNA Database (release 132)[53] available in the Kraken 2 web[54]. After assigning taxonomic labels to sequence readings, the Operational Taxonomic Units (OTUs) table was extracted using Pavian version[55].

*Biodiversity and clustering*. Taxonomic information of the samples with the abundance data for each OTUs was evaluated cross-sectionally and longitudinally. Alpha diversity metrics were computed using the R package *vegan* (functions *diversity, estimate* and *specnumber* for Shannon indicator, Chao1 index and observed richness, respectively). Beta diversity was assessed using weighted and unweighted Unifrac distances[56] (R package *phyloseq*, function *UniFrac*)[57]. We calculated the distances from each participant to its respective randomized donor, both in the FMT and placebo arm. Principal Coordinates Analysis (PCoA) of the abundance OTUs data was performed using the built-in R package *ape*, function *pcoa*. The first two dimensions of this PCoA were plotted separately for FMT arm and for the placebo arm in a graphics interchange format (GIF) representing the dimensions of each arm at each timepoint. The R package *vegan* (function *betadisper*) was used for the analysis of multivariate homogeneity of group dispersions. Biodiversity metrics were estimated considering all the taxonomic ranks with the exception of species level. This decision was made because this taxonomy rank is conventionally considered inaccurate. A Linear discriminant analysis (LDA) effect size (LEfSe) analysis was performed at the genus level from baseline to each timepoint to identify the OTUs more consistently engrafted in recipients across study timepoints (*lefse wrapper function*, *yingtools2* R package using the abundance OTUs tables). OTUs with LDA scores >2 were plotted in heatmaps (*pheatmap* function v1.0.12 with a hierarchical clustering in rows based on euclidean distances, R package).

**Measurements in blood samples**. A sample of fasting venous blood was obtained to determine the concentrations of glucose, total cholesterol, high-density lipoprotein cholesterol, and triglycerides using standard enzymatic methods. Plasma HIV RNA was measured using the Cobas Taq-Man HIV-1 assay (Roche Diagnostics Systems, Inc., Branchburg, NJ, USA). Cryopreserved plasma was assessed by immunoassay in triplicate for plasma levels of the inflammatory markers C-reactive protein C-reactive protein (CRP) (DCRP00, Quantikine ELISA kit, R & D Systems, Minneapolis, MN, USA) (CRP) (Labor Diagnostika, Nordhorn, Germany), sTNFR-II (DRT200, R&D Systems, Bio-Techne Corporation, Minneapolis, MN, USA), IP-10 (DIP100, R&D Systems, Bio-Techne Corporation, Minneapolis, MN, USA), sCD14 (AbClonal, Wuhan, China), sCD163 (AbClonal, Wuhan, China), LTA (Abbexa, Cambridge, UK), LBP (Boster Biological Technology, Wuhan, China), FABP2/IFABP (Boster Biological Technology, Wuhan, China D-dimers (Ray Biotech, Norcross, GA, USA),) according to the manufacturers' recommendations. For T-cell immunophenotyping, peripheral blood mononuclear cells (PBMCs) were isolated by Ficoll-Hypaque gradient-centrifugation and immediately stored in liquid nitrogen. T-cell immunophenotyping from thawed PBMCs was performed with the following antibody combination: CD3-VioBlue, CD4-Fluorescein isothiocyanate (FITC), CD8-VioGreen, CD28-Phycoerythrin (PE), CD38-APC and HLA-DR-APC-Vio770, and PD-1 (PD-1-PE-Vio770). Antibodies were purchased from Myltenyi Biotec (Bergisch Gladbach, Germany), and isotype controls were carried out. Briefly, PBMCs were incubated with the antibodies for 20 min at 4 °C, washed and resuspended in PBS containing 1% azide. Cells were analyzed using a Gallios flow cytometer (Beckman-Coulter, CA, USA). After initially gating lymphocytes according to morphological parameters, at least 30000 CD3 + T-cells were gated for each sample and analyzed with Kaluza software (Beckman-Coulter) (Supplementary Figure 10).

**Statistical methods**. Qualitative variables were reported as a frequency distribution whereas quantitative variables were described as medians with their interquartile ranges (IQRs). Since the distribution of all the assessed variables departed

from normality after the Shapiro Wilk tests, we used the Mann-Whitney U test for the between-group comparisons of continuous variables and Wilcoxon signed-rank matched-pairs test to evaluate differences in numerical outcomes between time-points. To compare the trajectories of numerical outcomes between treatment groups, we used linear mixed models which included the interaction term time-versus-treatment group as a fixed-effect and a random effect for each patient. To allow for correlations caused by repeated observations, we set an autoregressive covariance structure. When the observed trajectories were not linear, we used piecewise linear mixed models to compare the slope parameters between treatment groups at the intervals defined by the inflection points. A robust variance estimator was used given the deviations from normality. Continuous outcome variables were log-transformed when necessary to satisfy model assumptions. Because of the small sample sizes, no statistical comparisons were performed for *post-hoc* subgroup analyses. A two-sided $P$ value <0.05 was considered statistically significant. Statistical analysis was performed using Stata v16.0 (StataCorp LP, College Station, TX). The study investigators had access to the study data and approved the final version of the manuscript.

**Reporting summary**. Further information on research design is available in the Nature Research Reporting Summary linked to this article.

## Data availability

All the sequences are publicly available in the European Nucleotide Archive database under accession numbers PRJEB36786 (ERP120017). Participant's metadata are displayed in the supplemental table. Source data are provided with this paper.

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

## Acknowledgements

We thank all the study participants who contributed to this work, subjects who helped to disseminate the crowdfunding campaign to raise funds for this project, those who altruistically contributed with donations, as well as the clinical research staff who made this research possible. We especially thank Laura Luna and María Helena Álvarez for their technical support throughout the study. This work was supported by the Instituto de Salud Carlos III (Plan Estatal de I + D + i 2013–2016, project PI18/00154, a Gilead Fellowship (GLD16-00030), the SPANISH AIDS Research Network RD16/0025/0001project), and co-financed by the European Development Regional Fund 'A way to achieve Europe' (ERDF). The present investigation was also funded by the Instituto de Salud Carlos III and the Fundación Asociación Española contra el Cáncer within the ERANET TRANSCAN-2 program, grant number AC17/00022, a crowdfunding project from the precipita platform of the Fundación Española para la Ciencia y la Tecnología (FECYT) and a restricted grant from Finch Therapeutics. The SEIMC-GESIDA Foundation supported this study with safety and data monitoring (GESIDA 9116). The funder of the study had no role in study design, data collection, data analysis, data interpretation, or writing of the report. The corresponding author had full access to all the data in the study and had final responsibility for the decision to submit for publication.

## Author contributions

S.S.-V., M.O., S.M. study conceptualization; S.S.-V., J.A.P.-M., F.D., MJ.V., R.R., J.M.-S., and S.H. recruitment and clinical follow-up; U.A. handling of clinical specimens and delivery of the intervention; R.E. donor selection and shipment of FMT capsules from Openbiome to Hospital Universitario Ramón y Cajal; B.N. and M.M. dietary assessment and analysis; N.M., A.V. and C.G. flow cytometry and measurement of plasma bio-markers; S.S.-V. M.J.G. and A.M. supervision of the 16 S RNA gene sequencing experiments; S.S.-V statistical analysis; A.T., S.B., and V.L. bioinformatic analyses; J.Z. supervision of statistical analysis; S.S.-V. writing of the first version of the manuscript. All the authors reviewed and approved the manuscript.

## Competing interests

Outside the submitted work, S.S.-V. reports personal fees from ViiV Healthcare, Janssen Cilag, Gilead Sciences, and MSD as well as non-financial support from ViiV Healthcare and Gilead Sciences and research grants from MSD and Gilead Sciences. J.M.-S. reports non-financial support from ViiV Healthcare, Gilead Sciences, and Jannsen Cilag. J.P. reports grants, personal fees and non-financial support from ViiV Healthcare; grants, personal fees, and non-financial support from Gilead Sciences; grants, personal fees, and non-financial support from Janssen Cilag; personal fees from MSD; and personal fees from Abbvie. F.D. reports personal fees from Gilead. J.M.S. reports personal fees from ViiV Healthcare, Janssen Cilag, and Gilead Sciences. MJ.V. reports personal fees from Gilead Sciences, non-financial support from ViiV Healthcare, and Gilead Sciences, and grants from ViiV Healthcare. S.M. reports personal fees and non-financial from ViiV Healthcare, Janssen, Gilead Sciences, and MSD, as well as grants from MSD, ViiV Healthcare, and Gilead Sciences. S.B. and M.O. work for OpenBiome. R.E. was an employee at OpenBiome during the conduct of this study. A.T.-R., M.J.G., N.M., B.N., V.L., A.V., J.Z., C.G., M.M., R.R., S.H., U.A., and A.M. report no conflicts of interest.
