## [Peer Review File · Nature Communications]

REVIEWER COMMENTS

Reviewer #1 (Remarks to the Author):

The study by Serrano-Villar and co-authors encompasses a fecal microbiota transplant trial in people with HIV, a population that experiences microbial dysbiosis with previously reported links to markers of inflammation that are prognostic of morbidity. The authors find that repeated doses of FMT have an impact on the microbiome and a re-structuring of the community toward the donors. They also report improvements of IFAB-P, a marker of gut health, suggesting that FMT may improve GI barrier function which would be of great interest to the field.

Comments:

Please indicate the formulas used for mixed models across the study (including Figure 1), with clear indications of which variables were treated as fixed and random effects.

The motion graphic (GIF) depicting FMT recipient and placebo longitudinal shifts in microbiota should be performed using Principal Coordinates Analysis using beta diversity measures and not Principal Components Analysis. Furthermore, the states conclusion from this graphic ("Microbiome shifts were more pronounced in the FMT intervention than in the placebo group"), can be tested using the betadisper algorithm (and I believe should).

Apart from reference to Figure 2A, none of the statements in lines 339-361 have statistical or quantitative measures to support them. Please report this for all statements in this paragraph as they are quite important to the conclusions of the study.

The legend of Figure 2 states that black dots represent individual measurements, but there is only one dot per time point when there are several subjects per time point. Please show the actual individual measurements. This will be especially informative given the wide error bars on several of the measurements. Heterogeneity is to be expected, and it will help for it to be represented.

2 of the 4 subjects that received antibiotics prior to FMT did not show any discernible shift in composition toward the donor. This argues against the idea antibiotics were particularly helpful for engraftment, as only 2 subjects follow this trend (a number that may be the result of a spurious

observation), and the statement in the abstract is thus somewhat misleading: “Beta-diversity changes indicated mild

engraftment of donor’s and greater engraftment among the four subjects who had received antibiotics in the 12-week period before FMT.”

There is wide heterogeneity in the IFAB-P measurements for some subjects as indicated by confidence intervals in Figure 6H-J. Please show individual points for subjects by Donor in supplemental information, as was done in Figure 3. Also, what was the statistical result when comparing all post-FMT time points to the baseline for IFAB-P? Only results from three time points are shown. If this is not significant, the abstract, results, and discussion should be softened with regards to this observation.

Reviewer #2 (Remarks to the Author):

In this manuscript, Serrano-Villar et al. describe a randomized placebo-controlled trial of fecal microbiota transplant (FMT) in 30 people with HIV (PWH) on antiretroviral therapy (ART), with 15 in the placebo arm and 15 in the treatment arm. The authors administered FMT from three separate donors to each of 5 patients by capsule once weekly for 8 weeks. Stool and blood samples were collected for analysis for up to 48 weeks. Overall, the study demonstrates that FMT is generally well-tolerated in this population. They present evidence that the administration of FMT has an effect on the composition of the microbiome of the recipient. However, there are a number of issues related to the analysis, presentation and interpretation of the data. Below are points to be addressed:

- There are a number of issues that are not addressed related to the baseline differences between participants in the FMT and placebo groups. Those in the FMT group have been infected with HIV for much longer than those in the placebo group (19 years vs 5 years) and there are differences in nadir CD4 and CD4/CD8 T cell ratio. Additionally, 10 of 15 subjects from the FMT arm were exposed to antibiotics either before or during the study vs 3 of 15 in the placebo group (Table S1). It would be helpful to include detailed information about taxonomic composition and matched metadata and to use consistent patient number throughout the paper to better assess the potential impact of these differences. These differences should also be addressed in text with a discussion of the potential impact on the study.
- The differences in communities with FMT appear to be very small. The differences in alpha diversity measures are not significant overall and only reach significance when considering points <7 weeks. It is unclear why the analysis was done in this way and whether the “<7 week” timepoint includes the last FMT administration during week 7.

- The analysis of distance of communities from the donor to the recipient they present both weighted and unweighted unifracs distance from donor to recipient (Figure 2A). In the case of unweighted distance, the recipients in the treatment arm compared to the placebo. It is unclear what the comparison for the placebo arm is. For the placebo is this distance from baseline? The weighted unifracs comparison shows that the communities look more similar to the donor during the treatment period, which suggests that abundant OTUs from the donor are transferred to the recipient. But there is little resolution about what specific changes are occurring. Additionally, the analysis of patients in aggregate and over several time points appear to be confounded by auto-correlation between time points. All else equal, a patient should be self-similar between time points as measured by community distance metrics, meaning that a comparison over time should correct for this auto-correlation. A residual analysis can perhaps help with this auto-correlation, but the presented p-values are meaningless because of lumping patients and time points together in this way. One way to correct this issue would be to use a repeated measures ANOVA, but this approach is not robust to high levels of auto-correlation. In general, the justification for the statistical methods used is not present, and the authors need to provide a convincing argument and the assumptions of all statistical methods used to draw their conclusions. Overall, it seems the changes in community with FMT are modest at best and transient.
- Analysis of amplicon data is non-standard. Most of the field has moved to OTU-denoising approaches such as DADA2 (<https://doi.org/10.1038/nmeth.3869>) or deblur (DOI: 10.1128/mSystems.00191-16) to generate the community tables for downstream analysis.
- The observed difference in iFABP are overstated in the manuscript. There was a difference at one time point and it appears there was no correction for multiple comparisons.
- Overall the explanation of the rationale, detailed differences observed, and statistical methods used was sparse. The manuscript would benefit from additional editing for clarity, particularly in guiding the reader through the figures and data.

Additional minor comments:

Abstract:

line 29

ART Spell out anti-retroviral therapy the first time

Line 32

FMT spell out fecal microbiota transplant the first time it is used

Line 38

I-FABP spell out intestinal fatty-acid binding protein

The abstract is lacking information about the clinical outcomes that were measured – it only talks about changes in the microbiota and briefly states “attenuated HIV-associated dysbiosis”, but it seems these measures (and safety) would be the primary reasons for the study.

Introduction:

Lines 58-60:

Sentence starting with “As in any other...” seems superfluous; also unclear what the authors are wanting to say. I suggest removing

Line 68

Defined PWH again here (already defined in line 66)

Line 71:

Define synbiotic

The introduction is lacking a description of the aims and scientific questions in the remainder of the paper. These need to follow after Lines 83-86. For example “were there differences in HIV-associated dysbiosis in treatment and control?”

Methods

Lines 125-127

Please explicitly state what you mean by “the opposite microbiota signature”

Line 132:

What is the placebo?

Lines 147-148:

I think the “primary outcomes” should be moved to the abstract and the introduction. Until reading the methods, these were unclear.

Lines 198-199

Should probably say that “samples were stored in the stabilizer solution in Omnigene Gut Kits”

Lines 222-229

I think a more appropriate approach would be using DADA2 or some sort of denoising script to identify sequence variants within the dataset. I've only seen Kraken used in the context of WGS metagenomes for pulling out fragments of taxonomic markers... this approach is losing information that should be contained in the full-length amplicons. The downstream analysis should be done at the level of ASVs, not OTUs.

Results:

Throughout: OTU should say OTUs when plural

Figure 1: It seems data should be stratified by recipient because trends are no clear. Maybe the y-axes can be truncated too because there is a lot of unused space, which is making it harder to see the data points.

Figure S3A: Lines 332-335 are unclear. It looks like donor A and B lead to similar increases in diversity with donor C not doing much at all.

Figure 2: same comment as Figure 1... the data is too small to see because too much of the plot is empty; what is the "donor" in the placebo arm? Shouldn't this just be a comparison to self?

Figure 2B: there are few enough subjects that these should be stratified by patients instead of showing massive error bars.

Lines 354-355 It appears that donor A is noisier than donor B, and there is no statistical comparison being done here. (if there were, though, it would need to be stratified by patient and take into account the autocorrelation in the signal)

Line 358: confidence intervals are all overlapping zero, so it is hard to say something is "more apparent" here

Figure 3 is what I would prefer in place of Figure 2. There is no need for both. It's the same data, so I would suggest condensing the information into one figure.

Lines 364-366: Unclear how Figure 4 maps to what this sentence is saying

Figure 5. The legend needs more description of the components of the figure. What are the inputs into the calculations? How were recipients in the placebo arm allocated to donors if they didn't receive anything?

Discussion

Lines 410-413: the claim of a "significant and rapid decline" is overstated given the observed results; in all cases the confidence interval for treatment overlapped the placebo

Reviewer #3 (Remarks to the Author):

Summary

In this manuscript, the authors conducted a double-blind study, in which 30 HIV-infected subjects on ART with a CD4/CD8 ratio < 1 were randomized to either weekly fecal microbiota capsules or placebo for 8 weeks. FMT increased alpha diversity and beta-diversity indicated engraftment of donor. IFABP, a biomarker of intestinal damage that independently predicts mortality, was different between groups. The weighted B-diversity showed incremental engraftment without evidence of a plateau over time. This is novel and informative. The granular detail (time points, repeated dosing) is a testament to the detailed work conducted by the study team. The limitation of this study is the absence of a correlate with clinical benefit; the iFABP reduction is modest, associated with higher baseline value, and may suffer from multiple testing hypothesis.

Major comments

- Eligibility: how was use of probiotics handled?
- Stool donors: selected three donors in the higher quartile of fecal Bacteroides and Faecalibacterium genus abundance and butyrate concentrations, and in the lower quartile of Prevotella abundance.
- FMT dosage. How was induction with 10 capsules selected? For treatment of C diff infection, the dose is typically 30 capsule. The maintenance dose of 5 capsules per week is also lower than typical maintenance doses in other clinical trials.

- There is no mention of IFABP in the 'measurements in blood samples' section
- Antibiotic use occurred in 3 subjects in each group (beginning of results), but in Line 336, it states that 4 subjects received abtx. Please clarify. Abtx typically reduces alpha diversity and increases the distance in beta-diversity. Can the authors speculate why this was not observed in this study?
- For the beta diversity comparison to donor, what is the placebo group (figure 2a) being compared to?
- What was different about donor A with respect to impact on alpha diversity? Similarly, it appeared that there was no effect at all from donors B & C on beta-diversity (figure 2b).
- The IFABP data, particularly that derived from recipients of donor A, appear to decrease rapidly after the baseline value. However, the baseline values particularly from the recipients of Donor A are also higher at baseline. Therefore, is it possible that this is a random occurrence that the baseline value was high and there was a natural reversion to the mean? Is it possible to use several pre-baseline blood samples to calculate a mean baseline value?

Minor comments

- uLL should be uL (line 96)
- Richness increased from 250 to 287 vs. 252 to 254. The author mentions several alpha diversity metrics. They can include a brief sentence to describe the benefit of each approach (Chao, Shannon, Simpson).
- Outcomes mentioned in the methods included changes in CD4/8 T cell counts, CD4/8 ratio. GI symptom severity. Stool microbial outcomes, 11 systemic biomarkers. No mention of iFABP.

RESPONSE TO REVIEWERS

Manuscript number: NCOMMS-20-21964

Article title: Fecal microbiota transplantation in HIV: A pilot, randomized, placebo-controlled study

The authors would like to thank the Reviewers and Editors for their careful review of our manuscript, and for providing us with their very helpful comments and suggestions to improve the quality of the manuscript. The following responses have been prepared to address all of the referees' comments in a point-by-point fashion.

Reviewer #1 (Remarks to the Author):

The study by Serrano-Villar and co-authors encompasses a fecal microbiota transplant trial in people with HIV, a population that experiences microbial dysbiosis with previously reported links to markers of inflammation that are prognostic of morbidity. The authors find that repeated doses of FMT have an impact on the microbiome and a re-structuring of the community toward the donors. They also report improvements of IFAB-P, a marker of gut health, suggesting that FMT may improve GI barrier function which would be of great interest to the field.

Comments:

1. Please indicate the formulas used for mixed models across the study (including Figure 1), with clear indications of which variables were treated as fixed and random effects.

Author: All mixed models were fitted in a similar fashion for all analyses. To compare the linear trajectories between treatment groups, we included the interaction term time-versus-treatment group as a fixed-effect. Each patient was introduced as a random effect.

- To account for the autocorrelation caused by repeated observations that may result in optimistic estimates, we have set an autoregressive covariance structure, following the criticisms of Reviewer #2.
- We have revisited the whole statistical approach with a biostatistician (Javier Zamora), who has been added as a co-author.
- To approach the difficulty of the non-linear dynamics of many of the continuous variables assessed in the models, we have interrogated two time periods (weeks 0-8 and 12-48) using piecewise models, with similar settings.
- These changes have not substantially changed the effect estimates nor the main conclusions, although have produced more conservative estimates in some comparisons.

The following changes have been done following the comments of the Reviewers: We have modified the statistical methods to give consideration to this author concern (lines 372-378):

“To compare the trajectories of numerical outcomes between treatment groups, we used linear mixed models which included the interaction term time-versus-treatment group as a fixed-effect and a random effect for each patient. To allow for correlations caused by repeated observations, we set an autoregressive covariance structure. To compare the slope parameters between treatment groups at different time intervals, we used piecewise linear mixed models. A robust variance estimator was used given the deviations from normality. Continuous outcome variables were log-transformed when necessary to satisfy model assumptions.”

2. The motion graphic (GIF) depicting FMT recipient and placebo longitudinal shifts in microbiota should be performed using Principal Coordinates Analysis using beta diversity measures and not Principal Components Analysis. Furthermore, the states conclusion from this graphic (“Microbiome shifts were more pronounced in the FMT intervention than in the placebo group”), can be tested using the *betadisper* algorithm (and I believe should).

Authors: we appreciate the thoughtful comment. We have constructed a new motion graphic using the beta diversity measures represented in Principal Coordinates Analysis. We agree that this is a fairest approach, as avoid the zero-inflation issues generated by the Euclidean interpretation of the distances by Principal Coordinates Analyses. The updated GIF supports the statement referred by the Reviewer.

In addition, we have used the *betadisper* algorithm to represent the group dispersions (variances) and to test the homogeneity of variances. The differences between groups were significant (permuted P value =0.01), indicating that the beta diversity distances were more variable in the FMT group than in the placebo group.

This approach is complementary to the linear mixed modelling followed in the original manuscript (which accounts for time and repeated measurements), and has been included in the supplemental materials.

Figure 1. Principal Coordinates Analysis of beta diversity distances (Genus level) in study samples in each group. ANOVA for the between-group mean distances, $P = 0.0022$; permuted P value = 0.01.

3. Apart from reference to Figure 2A, none of the statements in lines 339-361 have statistical or quantitative measures to support them. Please report this for all statements in this paragraph as they are quite important to the conclusions of the study.

Authors: the statements reported now in lines 452-456 are supported by the animated figure representing the Principal Coordinates Analysis changes of the abundance OTU data and also by the new **figure S5** generated using *betadisper*, in which the observed differences were statistically significant. In addition, to make the text more self-explanatory the statements in this paragraph without duplicating the information shown in figure 2, we have included a measure of the size effect expressed as variable fold-change from baseline to the peak of effect, and included the p values for the comparison of slope parameters from baseline to week 8 and from week 12 to week 48.

“For unweighted Unifrac, which considers the presence/absence of taxa, the peak effect occurred at week 7 without evidence of a plateau (fold change, -4% in the FMT arm vs +1% in the placebo arm) and returned to baseline values after week 8 (FMT vs. placebo trajectories until week 8, $p=0.549$; beyond week 8, $p=0.829$). In contrast, for weighted Unifrac, which also considers the abundance of the taxa differentially abundant and their phylogenetic relatedness, the effect was incremental after each FMT, peaked at week 8 (fold change, -7% in the FMT arm vs +7% in the placebo arm; FMT vs. placebo trajectories until week 8, $p=0.029$; beyond week 8, $p=0.995$). These findings are indicative of a small and transient but detectable effect at the beta diversity level, and suggests that greater engraftment could have been achieved with additional FMT courses.”

4. The legend of Figure 2 states that black dots represent individual measurements, but there is only one dot per time point when there are several subjects per time point. Please show the actual individual measurements. This will be especially informative given the wide error bars on several of the measurements. Heterogeneity is to be expected, and it will help for it to be represented.

Authors: thank you for noting this. We have modified the figure 2 to show the value of each individual as a plot, superimposed to the mean and 95% confidence intervals.

5. 2 of the 4 subjects that received antibiotics prior to FMT did not show any discernible shift in composition toward the donor. This argues against the idea antibiotics were particularly helpful for engraftment, as only 2 subjects follow this trend (a number that may be the result of a spurious observation), and the statement in the abstract is thus somewhat misleading: “Beta-diversity changes indicated mild engraftment of donor’s and greater engraftment among the four subjects who had received antibiotics in the 12-week period before FMT.”

Authors: We agree with the Reviewer, the effect was largely determined by subject R2, who received 7 days of broad-spectrum antibiotics until the day before the FMT, and subject R1, who received 10 days of broad-spectrum antibiotics 14 weeks before the first FMT. In subjects R14 and R29. It must be noted that subject R14 had received azithromycin, a drug with a narrower antibiotic spectrum than that received by the other 3 patients (amoxicillin/clavulanate), and that the effect was particularly strong in subject R2, who basically stopped the antibiotics before starting the FMT.

To give consideration to this Reviewer’s comment, and because only 3 patients received antibiotics before the first (lead-in) FMT, we have modified the pertinent statements in the manuscript (Lines 472-504) in the main text, and we have reworded the mentioned

statement in the abstract (lines 40-41), in which we now simply indicate that the greatest engraftment was found in the participant who had received antibiotics most recently before FMT. In addition, we have truncated the Y axis in the figures to avoid blank space and facilitate the detection of patterns in the represented parameters.

6. There is wide heterogeneity in the IFAB-P measurements for some subjects as indicated by confidence intervals in Figure 6H-J. Please show individual points for subjects by Donor in supplemental information, as was done in Figure 3. Also, what was the statistical result when comparing all post-FMT time points to the baseline for IFAB-P? Only results from three time points are shown. If this is not significant, the abstract, results, and discussion should be softened with regards to this observation.

Authors: We compared the differences on biomarker trajectories between groups from baseline to each timepoint, but decided to represent in the figure the statistically significant p values for clarity. This analytical strategy differs from that of comparing the values at each visit with respect to baseline, which we have estimated using the same mixed model settings. The size effect is detailed in the contrast column.

	Contrast	Std. Err.	z	P> z	[95% Conf. Interval]	
log10 (IFABP)						
week@arm						
(1 vs 0) FMT	-.4007147	.2088643	-1.92	0.055	-.8100811	.0086518
(1 vs 0) Placebo	-.0215559	.0461346	-0.47	0.640	-.1119781	.0688663
(4 vs 0) FMT	-.3168359	.1544261	-2.05	0.040	-.6195055	-.0141662
(4 vs 0) Placebo	-.0065736	.0424356	-0.15	0.877	-.0897459	.0765987
(8 vs 0) FMT	-.2180037	.0946988	-2.30	0.021	-.40361	-.0323973
(8 vs 0) Placebo	-.2047227	.0960491	-2.13	0.033	-.3929754	-.01647
(24 vs 0) FMT	-.0862534	.1459671	-0.59	0.555	-.3723437	.1998368
(24 vs 0) Placebo	-.1417381	.1264203	-1.12	0.262	-.3895174	.1060411
(48 vs 0) FMT	-.5087771	.2049813	-2.48	0.013	-.9105329	-.1070212
(48 vs 0) Placebo	-.0823228	.1698634	-0.48	0.628	-.415249	.2506033

As it can be appreciated, there are significant decreases at 3 out of 5 follow-up visits in the FMT arm, and at 1 out of 5 visits in the placebo arm. The fact that the strongest decrease was found after the lead in FMT dose and that this remained significant until week 48 suggests that these findings were not due to spurious associations.

Even if this statistical approach yields more significant results than those reported from the comparison of treatment slopes, and because we had specified in the study protocol that we will report comparisons between slopes rather than cross-sectional comparisons of mean values between time-points, we did not include these results in the manuscript. Because this was a pilot study, our intention was to stay conservative and cautious in the statistical procedures.

As requested by the Reviewer, we have elaborated a new **Figure S9** showing the individual points, which has been added to the supplemental information. We have modified the legend of **figure 6** to make clear that the comparisons between treatment slopes from baseline and next time-points did not reach the threshold of statistical significance, except for the comparisons indicated in the IFABP subplots.

We have revised the statements regarding the significance of the IFABP changes.

-In the abstract and the results, the statements are essentially descriptive. In the abstract (lines 43-44), we only mention that statistical differences were detected. In the results (lines 523-527), we also refer to the magnitude of the reduction in each arm, with the respective P value of the comparison between slope parameters at each time interval.

-In the opening statements in the discussion that summarizes the main findings of the study, we have eliminated the term “significant” from the statement “*While the intervention did not cause any signal of immunological harm, a rapid decline of IFABP was observed in subjects receiving FMT compared to those receiving a placebo*” to soften it (line 536).

-In the discussion, section of limitations, we have included a statement to acknowledge that this association between FMT and greater IFABP decline must be interpreted cautiously, given the small sample sizes (lines 635-637).

“The main limitations are inherent to exploratory studies, such as a small sample size of 30 subjects. We measured 8 biomarkers of inflammation, and found differences between groups only in 1 of them, which could be due to spurious associations, given the small sample sizes assessed.”

Reviewer #2 (Remarks to the Author):

In this manuscript, Serrano-Villar et al. describe a randomized placebo-controlled trial of fecal microbiota transplant (FMT) in 30 people with HIV (PWH) on antiretroviral therapy (ART), with 15 in the placebo arm and 15 in the treatment arm. The authors administered FMT from three separate donors to each of 5 patients by capsule once weekly for 8 weeks. Stool and blood samples were collected for analysis for up to 48 weeks. Overall, the study demonstrates that FMT is generally well-tolerated in this population. They present evidence that the administration of FMT has an effect on the composition of the microbiome of the recipient. However, there are a number of issues related to the analysis, presentation and interpretation of the data. Below are points to be addressed:

1• There are a number of issues that are not addressed related to the baseline differences between participants in the FMT and placebo groups. Those in the FMT group have been infected with HIV for much longer than those in the placebo group (19 years vs 5 years) and there are differences in nadir CD4 and CD4/CD8 T cell ratio. Additionally, 10 of 15 subjects from the FMT arm were exposed to antibiotics either before or during the study vs 3 of 15 in the placebo group (Table S1). It would be helpful to include detailed information about taxonomic composition and matched metadata and to use consistent patient number throughout the paper to better assess the potential impact of these differences. These differences should also be addressed in text with a discussion of the potential impact on the study.

Authors: As the Reviewer points out, there was a statistically significant differences between case and controls in nadir CD4 ($p=0.015$), and years since HIV diagnosis ($p=0.06$), but the difference in baseline CD4/CD8 ratio differences were small ($p=0.383$). Please, note that the objective of the study was not to compare the microbiota of both groups or the effect of FMT in two different groups, but to understand the effects of FMT. The rationale to include a placebo arm was to have a sense of the stochastic changes in the microbiota and biomarkers measured over time, that would allow for a better interpretation of the findings in the FMT arm. Because controls received a placebo, the only reason by which these

differences could have confounded the results would be that the baseline study characteristics had affected the stochastic changes in the microbiota in the placebo arm. Besides there are no studies to support such an assumption, given the marked stability of the microbiota throughout the study, this seems unlikely. In figure 2, we represent the differences between subjects in the FMT (segregated by donor) vs. controls. The Prevotella/Bacteroides ratio was somewhat higher in the control group.

To assess the possible confounding effect of the variables significantly different between groups (nadir CD4, time since HIV diagnosis), we have introduced these two variables as fixed effects in the models more relevant for the results (i.e., those indicating changes in alpha diversity, unfrac distances and IFABP concentrations. The effects and P values of the interaction terms (treatment by term) were not affected. As an example, we have copied below the out from the model for alpha diversity changes (OTU counts):

*Mixed model for changes in OTU counts (without nadir CD4 and time since HIV diagnosis as fixed effects):

Log pseudolikelihood = -1000.4655	Wald chi2(3) = 39.28					
	Prob > chi2 = 0.0000					
(Std. Err. adjusted for 29 clusters in idnum)						

count	Coef.	Robust Std. Err.	z	P> z	[95% Conf. Interval]	

week	5.78024	.9247186	6.25	0.000	3.967825	7.592655
arm						
Placebo	-5.646132	30.85337	-0.18	0.855	-66.11763	54.82537
arm vs. week						
Placebo	-5.134038	2.04647	-2.51	0.012	-9.145046	-1.12303
constant	249.0011	19.72629	12.62	0.000	210.3383	287.6639

*Mixed model for changes in OTU counts (with nadir CD4 and time since HIV diagnosis as fixed effects):

Log pseudolikelihood = -998.34734	Wald chi2(5) = 53.84					
	Prob > chi2 = 0.0000					
(Std. Err. adjusted for 29 clusters in idnum)						

count	Coef.	Robust Std. Err.	z	P> z	[95% Conf. Interval]	

week	5.804279	.9244412	6.28	0.000	3.992408	7.616151
arm						
Placebo	-36.59395	27.1022	-1.35	0.177	-89.71328	16.52538
arm vs. week						
Placebo	-5.154945	2.044509	-2.52	0.012	-9.162109	-1.147782
Nadir CD4	.0343903	.1222277	0.28	0.778	-.2051717	.2739522

Yrs_sinceHIVdx	-3.068468	1.766362	-1.74	0.082	-6.530474	.3935385
constant	295.0028	43.9101	6.72	0.000	208.9406	381.065

Please, note that, in contrast with the Reviewer’s interpretation, the number of patients who received antibiotics either before or during the study was well balanced between arms: 8 in the FMT arm (id 1, 2 and 29, before starting the FMTs; and id 14, 18, 23, 24 and 30, during the follow-up) vs 7 in the placebo arm (subject 6, 8 and 21: 3 before the intervention; and subject 19, 25, 26 and 27, during the follow-up) in contrast with the Reviewer’s interpretation. The 2 extra instances correspond to subject #18, who received 3 different antibiotics for different infections. As we highlighted in blue in the table, only 3 patients in the FMT were exposed to antibiotics before the first FMT and 4 during the study. Please, note that the number of patients

Last, to further explore the between-group differences in the baseline microbiota composition we have performed a Principal Coordinates Analysis of beta diversity distances between the FMT and Placebo groups. ANOVA for the between-group mean distances, Adonis test, P value = 0.442.

Principal Coordinates Analysis (PCoA) of beta diversity distances between the FMT and placebo groups at baseline. ANOVA for the between-group mean distances, Adonis test, P value = 0.442.

Actions:

- We have included a statement in the footnote and in the results to avoid misinterpretation in the numbers.
- In the discussion, we have acknowledged as a limitation the fact that there were some differences in the baseline characteristics. We have also added a statement in the figure legends 1, 2, and 6 to express that the coefficients and P values reported remained unchanged after adjustment for nadir CD4 and time since HIV diagnosis.
- As requested by the Reviewer, we have added a new table in the supplemental methods with the individual metadata of the study participants.

2• The differences in communities with FMT appear to be very small. The differences in alpha diversity measures are not significant overall and only reach significance when considering points <7 weeks. It is unclear why the analysis was done in this way and whether the “<7 week” timepoint includes the last FMT administration during week 7.

Authors: Because this was a pilot study and multiple biomarkers are analysed, we believe that all the p values must be interpreted cautiously, as it is acknowledged in the manuscript. Conversely, non-significant comparisons could be due to lack of power. We believe that a careful interpretation of figures is more valuable than the p values. We endorse the American Statistician Association’s recommendation to move beyond $p < 0.05$, especially in studies with small sample sizes (Wasserstein and Lazar 2016; Amrhein, Greenland, and McShane 2019).

When the observed trajectories were not linear, we used piecewise linear mixed models to compare the slope parameters between treatment groups at the intervals defined by the inflection points. This is necessary when mixed models are used and the observed longitudinal data are not linear. The fact that introducing all time-points in the model decreased the statistical significance does not argue that earlier changes are significant, and is due to the model constraints (because mixed model assumed linear slopes, a negative slope in the early weeks will approach zero if the model includes later timepoints in which no effect was seen).

Please, note that during the review process, we have updated the unifracs distances accuracy by using the full topological tree from the silva last database update, which has resulted in some changes in the range of the Y axes of figures 2 and 3 that have not affected the interpretation of the results.

Action:

- We have reworded the methods to justify how the time intervals were selected.
- We have changed the Y axis in figure 2 to avoid blank space and facilitate the visualization of the different patterns in each group.
- We have modified the methods to clarify the mixed models’ specifications and how the time intervals analysed were defined (lines 344-351).

3• The analysis of distance of communities from the donor to the recipient they present both weighted and unweighted unifracs distance from donor to recipient (Figure 2A). In the case of unweighted distance, the recipients in the treatment arm compared to the placebo. It is unclear what the comparison for the placebo arm is. For the placebo is this distance from baseline?

Authors: We appreciate this thoughtful observation. The study participants were randomly assigned to active or placebo in blocks of three (one per donor) by a computer-generated randomized number system. This implies that subjects in the placebo arm had a donor assigned even if they received a placebo. To represent both groups under the same conditions, the Unifracs distances shown in figure 2 represent, in the case of participants in the placebo arm, the distance at each visit from their assigned donor. This was not well explained in the figure legend.

We have modified the legend of figures 3 and 5 to better explain how the Unifrac distances were calculated in the placebo arm.

4. The weighted unifrac comparison shows that the communities look more similar to the donor during the treatment period, which suggests that abundant OTUs from the donor are transferred to the recipient. But there is little resolution about what specific changes are occurring.

Comment: The results suggest that only a limited number of communities were consistently engrafted across study visits. The objective of figure 5 is to provide a better resolution of this statement. Figure 5 represents in both arms which OTU were engrafted, when, for how long and in which group, in relation to the taxa present in the FMT donors. We have clarified the legend of figure 5 to better explain what the figure represents.

To detail the specific changes occurring in the FMT arm, we have summarized in the text the taxa that were more consistently enriched over time in the FMT group (*Anaerostipes* spp., *Blautia* spp., *Dorea* spp., and *Fusicatenibacter* spp, Ruminococcaceae family, Lachnospiraceae family). To avoid populating the text with an exhaustive list of taxa that would affect the readability and to avoid duplicating the information across text and figures, we have not listed the full list of taxa significantly enriched in the FMT group, which can be identified in the figure 5.

5. Additionally, the analysis of patients in aggregate and over several time points appear to be confounded by auto-correlation between time points. All else equal, a patient should be self-similar between time points as measured by community distance metrics, meaning that a comparison over time should correct for this auto-correlation. A residual analysis can perhaps help with this auto-correlation, but the presented p-values are meaningless because of lumping patients and time points together in this way. One way to correct this issue would be to use a repeated measures ANOVA, but this approach is not robust to high levels of auto-correlation. In general, the justification for the statistical methods used is not present, and the authors need to provide a convincing argument and the assumptions of all statistical methods used to draw their conclusions. Overall, it seems the changes in community with FMT are modest at best and transient.

Author: we appreciate this technical comment. We agree, auto-correlation could result in too optimistic estimates. To avoid overfitting, the mixed models were included as a continuous variable, instead of a dummy variable, which would have generated 13 covariates. The covariance structure was set as independent.

Action: we have changed the models' specifications to set an autoregressive covariance structure, which corrects for autocorrelation, and we have repeated all the analyses reported and modified them through the text and figures. This approach has introduced minor modifications of the computed P values at the third decimal place and has not resulted in any change affecting the results and interpretations. The statistical approach has been supervised by a biostatistician (Javier Zamora), which has been included as a co-author. The pertinent changes have been made in the methods.

6• Analysis of amplicon data is non-standard. Most of the field has moved to OTU-denosing approaches such as DADA2 (<https://doi.org/10.1038/nmeth.3869>) or deblur (DOI: 10.1128/mSystems.00191-16) to generate the community tables for downstream analysis.

Authors: DADA2 is a model-based approach for correcting amplicon errors without constructing OTUs and is integrated in Qiime2 for trimming, denosing and chimera and PhiX removal (link). Kraken was released in 2014 and has been shown to provide exceptionally fast and accurate classification for shotgun metagenomics sequencing projects. Kraken2, which matches the accuracy and speed of Kraken 1, now supports 16S rRNA databases, allowing for direct comparisons to QIIME and similar systems. The direct comparison of Kraken2 with QIIME2 for both the Greengenes and SIVA databases demonstrated that Kraken2 outperforms QIIME2 in terms of computational requirements, runtime, and accuracy (PMID: 32859275).

To give consideration to this Reviewer's concern, we have compared the composition tables generated using DADA2 vs. Kraken2 from the 16SrRNA data generated from the study samples. The **figure 1** copied below represents the correlations between the different taxa obtained using DADA2 and Kraken2 tools at different taxonomic levels. The correlation between the results obtained from each tool was very high: Phylum level, Rho 0.92, $p < 0.001$; Class level, Rho 0.92, $p < 0.001$; Order level, Rho 0.92, $p < 0.001$; family level, Rho 0.90, $p < 0.001$; genus level, Rho 0.73, $p < 0.001$.

Figure 1. Correlations between the different taxa obtained using DADA2 and Kraken2 tools at different taxonomic levels

In the methods, we have added the reference of the work that compares Kraken2 and QUIIME2.

7• The observed difference in iFABP are overstated in the manuscript. There was a difference at one time point and it appears there was no correction for multiple comparisons.

Authors: we have emphasized throughout the text that this was a pilot study, the secondary outcomes including changes in the microbiota and the analysed biomarkers are exploratory, and avoided strong claims. Some of the bioinformatic approaches used (LEfSe analysis) imply corrections for multiple comparisons in highly dimensional datasets. As stated in the methods, we did not adjust for multiple comparisons. As it is widely accepted, simply describing what tests have been performed and why is generally the best way of dealing with multiple comparisons in pilot studies (Perneger, BMJ 1999).

We have revised the statements regarding the significance of the IFABP changes.

-In the abstract and the results, the statements are essentially descriptive. We have avoided referring to IFABP changes in the statement summarizing the main findings in lines 37-38:

“Fecal microbiota transplantation (FMT) was safe, not-related to severe adverse events, and attenuated HIV-associated dysbiosis”.

-In the abstract (line 44), we only mention that statistical differences were detected. In the results (lines 521-527), we also refer to the magnitude of the reduction in each arm, with the respective P value of the comparison between slope parameters at each time interval.

-In the opening statements in the discussion that summarizes the main findings of the study, we have eliminated the term “significant” from the statement *“While the intervention did not cause any signal of immunological harm, a rapid decline of IFABP was observed in subjects receiving FMT compared to those receiving a placebo”* to soften it (line 536).

-In the discussion, section of limitations, we have included a statement to acknowledge that this association between FMT and greater IFABP decline must be interpreted cautiously, given the small sample sizes (line 635-637).

“The main limitations are inherent to exploratory studies, such as a small sample size of 30 subjects. We measured 8 biomarkers of inflammation, and found differences between groups only in 1 of them, which could be due to spurious associations, given the small sample sizes assessed.”

8• Overall the explanation of the rationale, detailed differences observed, and statistical methods used was sparse. The manuscript would benefit from additional editing for clarity, particularly in guiding the reader through the figures and data.

Authors: In the introduction, we have reworded the statements in lines 89-100 to better explain the rationale. In the last paragraph, we have further elaborate on the rationale in the context of the study design.

We have reworded the text in lines 445-477 to facilitate the flow between the text and figures 3 and S3.

We have revised and extensively edited the figure legends to improve the clarity and provide a better guidance to the reader through figures.

Additional minor comments:

9. Abstract:

line 29. ART Spell out anti-retroviral therapy the first time. Line 32. FMT spell out fecal microbiota transplant the first time it is used. Line 38 . I-FABP spell out intestinal fatty-acid binding protein.

Authors: we have spelt out these acronyms, as indicated.

10. The abstract is lacking information about the clinical outcomes that were measured – it only talks about changes in the microbiota and briefly states “attenuated HIV-associated dysbiosis”, but it seems these measures (and safety) would be the primary reasons for the study.

Authors: as reported in the study protocol available at <https://clinicaltrials.gov/ct2/show/NCT03008941> and also attached to the submission, the principal outcome was safety. Secondary outcomes were exploratory and included changes in CD4+ T cell counts, CD8+ T cell counts, CD4/CD8 ratio, inflammatory markers, T cell activation and markers of enterocyte barrier function through week 48.

In the abstract we have now specified that the intervention was not related to severe adverse events, which was the main outcome. From the exploratory outcomes, given the space constraints for this section (150 words) we have mentioned only the positive findings.

Introduction:

11. Lines 58-60. Sentence starting with “As in any other...” seems superfluous; also unclear what the authors are wanting to say. I suggest removing

Authors: following this Reviewer’s suggestion, we have removed this statement.

12. Line 68, Defined PWH again here (already defined in line 66)

Authors: Thank you for noting this duplication; we have corrected it.

13. Line 71. Define synbiotic

Authors: We have included the definition of synbiotic.

14. The introduction is lacking a description of the aims and scientific questions in the remainder of the paper. These need to follow after Lines 83-86. For example “were there differences in HIV-associated dysbiosis in treatment and control?”

Authors: following this comment, in the last paragraph, we have further elaborate on the rationale in the context of the study design (lines 88-106).

Methods

15. Lines 125-127. Please explicitly state what you mean by “the opposite microbiota signature”

Authors: we have detailed in the methods that this refers to a microbiota characterized by enrichment for Prevotella genus and depletion of Bacteroides and Faecalibacterium genus.

16. Line 132. What is the placebo?

Authors: The placebo capsules contained glycerol, cocoa butter, and inert, non-toxic brown pigment (in place of stool). They were produced using the same protocol as active capsules.
Actions: We have included this information in the methods (lines 163-165).

17. Lines 147-148. I think the “primary outcomes” should be moved to the abstract and the introduction. Until reading the methods, these were unclear.

Authors: Following this suggestion, we have emphasized in the abstract the safety results and detailed in the introduction the primary and secondary outcomes (lines 102-106). It is challenging to describe in more detail the study design in the abstract (150 words) without removing some of the results reported.

18. Lines 198-199. Should probably say that “samples were stored in the stabilizer solution in Omnigene Gut Kits”

Authors: Actually, the Omnigene Gut kit is a device specifically designed for stool collection for microbiome analysis that also contains a stabilizer solution. For clarity, we have added that the stabilizer is a solution.

19. Lines 222-229. I think a more appropriate approach would be using DADA2 or some sort of denoising script to identify sequence variants within the dataset. I’ve only seen Kraken used in the context of WGS metagenomes for pulling out fragments of taxonomic markers... this approach is losing information that should be contained in the full-length amplicons. The downstream analysis should be done at the level of ASVs, not OTUs.

Authors: Please, see the related response to comment #6. While Kraken1 was developed to provide faster and more accurate classification for shotgun metagenomic data, Kraken2 supports 16S rRNA data.

Results:

20. Throughout: OTU should say OTUs when plural

Authors: This change has been performed.

21. Figure 1: It seems data should be stratified by recipient because trends are no clear. Maybe the y-axes can be truncated too because there is a lot of unused space, which is making it harder to see the data points.

Authors: Following this suggestion, we have modified the Y axis across figures to avoid blank space and to facilitate the visualization of different patterns between groups.

22. Figure S3A: Lines 332-335 are unclear. It looks like donor A and B lead to similar increases in diversity with donor C not doing much at all.

Authors: Following a previous suggestion from this Reviewer, we have truncated the Y axis to facilitate the observation of patterns. As can be now more easily appreciated, from the three alpha diversity metrics, the observed counts increased in donor A and C (more clearly in C), Chao 1 increased in all 3 donors (more clearly in C), and Shannon increased in all 3 donors (more clearly in A).

We have reworded the statement that was not accurate enough, to state that the largest improvement of alpha diversity was found for donor A as measured by Shannon index (lines 403-406).

23. Figure 2: same comment as Figure 1... the data is too small to see because too much of the plot is empty; what is the “donor” in the placebo arm? Shouldn’t this just be a comparison to self?

Authors: Following a previous suggestion from this Reviewer, we have truncated the Y axis to facilitate the observation of patterns. The Unifrac distances in the placebo group were calculated from each sample to the assigned donor, in order to understand the similarities with donors explained by randomness.

24. Figure 2B: there are few enough subjects that these should be stratified by patients instead of showing massive error bars.

Authors: We understand the Reviewer’s point. We have provided the independent trajectories as a figure in the main text (**figure 3**). We believe that both representations are complementary. While figure 2 facilitates the recognition of patterns, figure 3 allows a detailed evaluation of each subject and puts in context the timing of the antibiotic.

25. Lines 354-355 It appears that donor A is noisier than donor B, and there is no statistical comparison being done here. (if there were, though, it would need to be stratified by patient and take into account the autocorrelation in the signal)

Authors: Because of the small sample sizes, we deliberately avoided reporting statistical comparisons in all *post hoc* subgroup analyses (i.e., analysis by donor and by use of antibiotics).

We have performed the statistical comparisons requested by the Reviewer, following the same modelling strategy described above.

Time interval: Week 0-8		Wald chi2(7)				
= 14.57						
Log pseudolikelihood = 667.89069		Prob > chi2		= 0.0419		
(Std. Err. adjusted for 29 clusters in idnum)						
unifrac_w	Coef.	Robust Std. Err.	z	P> z	[95% Conf. Interval]	
week	-.0000937	.0003184	-0.29	0.769	-.0007178	.0005305
arm						
(ref. = placebo)						
FMT, A	-.0439233	.0242916	-1.81	0.071	-.091534	.0036874
FMT, B	-.0019481	.0129536	-0.15	0.880	-.0273367	.0234404
FMT, C	-.0067704	.0213992	-0.32	0.752	-.048712	.0351712
arm by week						
FMT, A	-.0014555	.0006522	-2.23	0.026	-.0027338	-.0001773
FMT, B	-.0016504	.0007873	-2.10	0.036	-.0031934	-.0001073
FMT, C	-.0003921	.001073	-0.37	0.715	-.0024952	.0017109
constant	.17583	.0104219	16.87	0.000	.1554035	.1962565

The analysis indicates that the differences in Unifrac distances of subjects in the FMT group for donors A and B, were statistically significant.

We had included these P values in the graph, but, since this was a subanalysis of small sample sizes (N=5), have explicitly avoided mentioning them in the text to avoid overstating their relevance.

26. Line 358: confidence intervals are all overlapping zero, so it is hard to say something is “more apparent” here

Authors: we agree with the Reviewer that there is no clearly different pattern at the alpha diversity level. The pattern of mean changes at the beta diversity level appears different between groups, and the wideness of the confidence intervals in the antibiotic group is related to the very small sample size in this group. Note that we deliberately used the term “appears” because this verb refers to “give an impression”.

Action: we have modified the statement referred by the Reviewer to specify that the visual inspection of the beta diversity plots was suggestive of changes in the mean values observed in the antibiotic group (Lines 403-406).

27. Figure 3 is what I would prefer in place of Figure 2. There is no need for both. It’s the same data, so I would suggest condensing the information into one figure.

Authors: We have provided the independent trajectories as a figure in the main text (**figure 3**). We believe that both representations are complementary. Figure 2 facilitates the recognition of patterns and represents and is necessary to understand the effect sizes of the differences for which statistical comparisons are reported. Figure 3 allows a detailed evaluation of each subject and puts in context the timing of the antibiotic, but we think that eliminating figure 2 will impact negatively the interpretability of the results. Still, we can move any figure to the supplemental materials if the Reviewer thinks that this is something necessary,

28. Lines 364-366: Unclear how Figure 4 maps to what this sentence is saying

Authors: we have reworded this sentence for clarity (line 480-482).

29. Figure 5. The legend needs more description of the components of the figure. What are the inputs into the calculations? How were recipients in the placebo arm allocated to donors if they didn’t receive anything?

Authors: This heatmap represents the LDA distances at the genus level from study participants to donors at each time point in the placebo and FMT groups. For the LDA distances calculations in the participants allocated to receive a placebo, the values represented were calculated as the distances from each timepoint to the assigned donor. Each column represents one time point. The three columns represented on the left of the pale-yellow cells represent the distance from the assigned donor to the baseline visit, and inform on the relative abundance of the represented genus on study participants with respect to donors at baseline.

For clarity, we have modified the legend of figure 5.

30. Discussion. Lines 410-413: the claim of a “significant and rapid decline” is overstated given the observed results; in all cases the confidence interval for treatment overlapped the placebo.

Authors: we agree with the Reviewer. We have modified this statement. Instead of “a significant and rapid decline”, the text says “an early decline” (line 536).

Reviewer #3 (Remarks to the Author):

Summary

In this manuscript, the authors conducted a double-blind study, in which 30 HIV-infected subjects on ART with a CD4/CD8 ratio <1 were randomized to either weekly fecal microbiota capsules or placebo for 8 weeks. FMT increased alpha diversity and beta-diversity indicated engraftment of donor. IFABP, a biomarker of intestinal damage that independently predicts mortality, was different between groups. The weighted B-diversity showed incremental engraftment without evidence of a plateau over time. This is novel and informative. The granular detail (time points, repeated dosing) is a testament to the detailed work conducted by the study team. The limitation of this study is the absence of a correlate with clinical benefit; the iFABP reduction is modest, associated with higher baseline value, and may suffer from multiple testing hypothesis.

Authors: We appreciate the overall positive assessment. Given the knowledge gaps in the framework of the study, we believe that the study is highly informative because we do not see any signal of safety concerns and, beyond, the data suggest possible beneficial effects, although we acknowledge that the magnitude of the IFABP improvement was not impressive. The main contribution of this study is that it will encourage the field to further explore this intervention searching for a stronger effect, for example, by increasing the FMT dose, sample sizes and using preconditioning antibiotic treatment.

Major comments

1• Eligibility: how was use of probiotics handled?

Authors: Probiotic use was collected as part of the dietary assessment, but no participant used specific probiotic supplementation beyond the probiotics that could be naturally present in yogurts. A previous placebo-controlled study from our group in HIV-infected individuals in which we specifically addressed the question whether a 48-week supplementation with probiotic and prebiotics could affect immune parameters and the gut microbiota composition in HIV and immune parameters essentially found no effect (PMID: 29788075). Actually, this lack of effect after probiotic supplementation prompted us to explore this FMT intervention, for which we anticipated that it could result in a stronger effect. For all these reasons, we did not elaborate on probiotic use in the manuscript.

2• Stool donors: selected three donors in the higher quartile of fecal Bacteroides and Faecalibacterium genus abundance and butyrate concentrations, and in the lower quartile of Prevotella abundance.

Authors: This donor's microbiota profile has been summarized in the methods and in the discussion (lines 160-165; 563-566).

3• FMT dosage. How was induction with 10 capsules selected? For treatment of C diff infection, the dose is typically 30 capsule. The maintenance dose of 5 capsules per week is also lower than typical maintenance doses in other clinical trials.

Authors: This decision was taken with the premise that the main study objective was to assess safety. Because of the safety concerns inherent to explore FMT safety in immunocompromised patients, we reasoned that it would be safe to reduce the lead in dose typically used in *Clostridioides difficile* FMT trials (30 capsules, as the Reviewer points out).

At the time of the study conception, there were no similar trials to the best of our knowledge which could have informed us to decide the consecutive doses. Hence, because of the safety primary outcome, we assumed that we would administer a rather cumulative low FMT dose over the study period. We feel that the finding that we were able to detect differences between groups even with this dosing schedule and without antibiotic pre-conditioning treatment will encourage other groups to further explore this intervention.

In the methods, we have further elaborated on the rationale to select the dosage (lines 175-177).

4• There is no mention of IFABP in the ‘measurements in blood samples’ section

Authors: We had omitted mentioning IFABP in the list of outcomes in blood. The ELISA kit manufacturer for IFABP was mentioned in the methods (line 323).

We have listed IFABP as the marker of enterocyte integrity measured as an outcome (line 198).

5• Antibiotic use occurred in 3 subjects in each group (beginning of results), but in Line 336, it states that 4 subjects received abtx. Please clarify. Abtx typically reduces alpha diversity and increases the distance in beta-diversity. Can the authors speculate why this was not observed in this study?

Authors: Thank you for noting this inconsistency. We have now clarified in lines 447-477, that donor's microbiota engraftment on participants in the FMT arm was more apparent in the 4 subjects who were exposed to antibiotics either before (N=3) or early after the first FMT (N=1) at both alpha (Figure S3B) and beta (Figure S3C) diversity levels. Because this study was not designed to assess the effects of antibiotic use but we had to deal with antibiotic exposure to interpret the results, it is challenging to draw firm conclusions on this regard. With respect to the lack of effect on alpha and beta diversity typically associated with antibiotic exposure, we think that this can be explained by i) the heterogeneity of the antibiotic used. Note that most studies assessing the impact of antibiotics used broad-spectrum combinations, such as vancomycin + metronidazole. Because here antibiotic used reflected real life use, different antimicrobials were prescribed, including some with presumably a weak impact on gut microbiota (penicillin, azithromycin, fluconazole or nitazoxanide), ii) most antibiotic exposures occurred beyond 1 week before the sample collection, which could have allowed for a recovery of the previous microbiota configuration. The compositional barplots shown in figure 4 strongly argue that the microbiota composition of the most abundant genus was stable in most subjects throughout the study.

6• For the beta diversity comparison to donor, what is the placebo group (figure 2a) being compared to?

Authors: Please, see response #3 to Reviewer #2.

7• What was different about donor A with respect to impact on alpha diversity? Similarly, it appeared that there was no effect at all from donors B & C on beta-diversity (figure 2b).

Authors: Because all three donors were selected based on a similar microbiota profile, we think that the more prominent differences observed in participants who received FMT from

this donor are more related to the fact that 3 out of 5 recipients had received antibiotics before FMT. Changes were especially drastic in recipient #2, who had stopped antibiotics the day before FMT. Yet anecdotal, we believe that these findings support the notion that preconditioning antibiotic treatment is important for a better donor's microbiota engraftment in recipients, which is an open question in the field.

This point has been discussed in lines 637-642:

“Our study was not designed to study the effect of antibiotic preconditioning treatment, and the antibiotics received by the 4 subjects near the first FMT administrations included different antibiotics administered during a wide time frame. Hence, our findings regarding the effects of antibiotic treatment on donor's microbiota engraftment on participants must be interpreted with caution.”

8• The IFABP data, particularly that derived from recipients of donor A, appear to decrease rapidly after the baseline value. However, the baseline values particularly from the recipients of Donor A are also higher at baseline. Therefore, is it possible that this is a random occurrence that the baseline value was high and there was a natural reversion to the mean? Is it possible to use several pre-baseline blood samples to calculate a mean baseline value?

Authors: We appreciate this well-taken comment. Unfortunately, we do not have pre-baseline plasma samples to address this question. The baseline confidence intervals of the three subgroups of patients in the FMT overlapped, which at least indicates that the between-group differences were not significant.

Minor comments

9• uLL should be uL (line 96)

Authors: Thank you for spotting this typo, that we have corrected.

10• Richness increased from 250 to 287 vs. 252 to 254. The author mentions several alpha diversity metrics. They can include a brief sentence to describe the benefit of each approach (Chao, Shannon, Simpson).

Authors: We agree that this suggestion will help the reader to better interpret the results. We have added a brief description of the alpha diversity metrics reported in the results (lines 401-403).

11• Outcomes mentioned in the methods included changes in CD4/8 T cell counts, CD4/8 ratio. GI symptom severity. Stool microbial outcomes, 11 systemic biomarkers. No mention of iFABP.

Authors: Thank you for noting this. In the methods, we have listed IFABP as the outcome of gut epithelial damage analyzed in the study.

REVIEWER COMMENTS

Reviewer #1 (Remarks to the Author):

The authors have satisfactorily addressed my comments.

I was asked to comment on the author responses to Reviewer 2. I think they generally addressed the reviewer's concerns, with regards to auto-correlation and clarifications in text. Looking more deeply, I did have the following critiques of the rebuttal based on Reviewer 2's comments, below:

[in response to the rebuttal to reviewer 2's point #2]:The majority of microbiome differences appear to be isolated to the period during which capsules were administered (only Chao indices of all the measures including beta and alpha diversity measures were significant past week 6). Low-abundance food-borne bacteria such as those in cheese are known to be observable in the feces after consumption, though do not engraft durably and likely do not impact the host. Thus, the fact the changes are significant primarily during capsule intake may reflect detection of capsule bacteria that may be biologically inert and do not to engraft durably (Figure 2) and thus may not greatly impact the host. Please consider mentioning this in the abstract (i.e. "Beta-diversity changes indicated mild engraftment of donor's microbiota during the treatment period". Lines 36-37) as well as discussing this possible interpretation in the Discussion, as this may also explain apparent beta diversity distances to donor only during the FMT treatment period.

[in response to the rebuttal to reviewer 2's point #7]: It should likely be included in line 35 that dysbiosis was transiently attenuated. To say simply that it was attenuated may be misinterpreted to imply durable, long-lasting attenuation.

Reviewer #3 (Remarks to the Author):

This reviewer appreciates the detailed response from the corresponding author and study team. Several concerns and questions remain.

- Can the author confirm that OpenBiome had available butyrate concentrates by donors and that the butyrate level was used to select the 3 donors (line 157)? As such, were measurements of butyrate performed in the recipients before and after FMT?
- Given the number of tested outcomes (multiple inflammatory markers, markers of microbial translocation, T-cell activation markers, and markers of gut epithelial injury), it seems that any evidence will need to be adjusted by a statistical correction e.g., Bonferroni or a less strict alternative. Alternatively, the result must be interpreted with caution.
- What is the relationship between the patients who experienced the most engraftment and their change in levels of I-FABP?
- Would the significance observed in IFABP over time persist if a Wilcoxon signed-rank matched pair test was used? This reduces the effect of outliers e.g., Mann-Whitney.
- This reviewer is concerned with the overall conclusion that 'significant differences between groups were observed in IFABP', when the results suggest that the difference may have been driven by few individuals (recipients of donor A) with high baseline pre-FMT IFABP levels (figure 6J). If one or two of those individuals were removed from the analysis, would the significance persist? In addition, was the delta IFABP correlated to the delta B-diversity? Consider a more conditional statement if the distribution of change in IFABP is not even across study participants.

RESPONSE TO REVIEWERS

Manuscript number: NCOMMS-20-21964A

Article title: Fecal microbiota transplantation in HIV: A pilot, randomized, placebo-controlled study

The authors would like to thank the Reviewers and Editors for their careful review of our manuscript, and for providing us with their very helpful comments and suggestions to improve the quality of the manuscript. The following responses have been prepared to address all of the referees' comments in a point-by-point fashion.

REVIEWER COMMENTS

Reviewer #1 (Remarks to the Author):

The authors have satisfactorily addressed my comments.

I was asked to comment on the author responses to Reviewer 2. I think they generally addressed the reviewer's concerns, with regards to auto-correlation and clarifications in text. Looking more deeply, I did have the following critiques of the rebuttal based on Reviewer 2's comments, below:

[in response to the rebuttal to reviewer 2's point #2]:The majority of microbiome differences appear to be isolated to the period during which capsules were administered (only Chao indices of all the measures including beta and alpha diversity measures were significant past week 6). Low-abundance food-borne bacteria such as those in cheese are known to be observable in the feces after consumption, though do not engraft durably and likely do not impact the host. Thus, the fact the changes are significant primarily during capsule intake may reflect detection of capsule bacteria that may be biologically inert and do not to engraft durably (Figure 2) and thus may not greatly impact the host. Please consider mentioning this in the abstract (i.e. "Beta-diversity changes indicated mild engraftment of donor's microbiota during the treatment period". Lines 36-37) as well as discussing this possible interpretation in the Discussion, as this may also explain apparent beta diversity distances to donor only during the FMT treatment period.

Authors: Following the Reviewer's advice, we have emphasized in the abstract that "*Beta-diversity changes indicated mild engraftment of donor's microbiota during the treatment period*" (line 38).

We have further addressed the potential implications in the discussion (lines 503-508): "*Unifrac distances analysis from baseline to donors indicated that incremental engraftment of donor's microbiota on recipients occurred, without a clear threshold effect, suggesting that stronger engraftment could have been achieved if additional FMT had been given. Besides, this effect was limited to the treatment period, indicating that boosting doses may be required to*

maintain long-lasting effects at the beta diversity level. This observation is also relevant to the design of future studies in this direction.”

[in response to the rebuttal to reviewer 2’s point #7]: It should likely be included in line 35 that dysbiosis was transiently attenuated. To say simply that it was attenuated may be misinterpreted to imply durable, long-lasting attenuation.

Authors: While the strongest effects were detected during the period in which capsules were administered, we detected the following long-lasting effects (present at week 48) in the FMT arm that were not appreciated in the placebo arm:

1. Alpha Diversity: a significant increase of alpha diversity (observed OTU) at week 48 ($p=0.037$)
2. A higher number of taxa driving significant differences at week 48 with respect to baseline in the FMT group (16 taxa) than in the placebo group (3 taxa).
3. A significant engraftment of Ruminococcaceae and Lachnospiraceae families: these families include the major butyrate producers, which depletion is the main hallmark of HIV-associated dysbiosis (Vujkovic-Cvijin & Somsouk, 2019). The fact that the FMT donors were selected based on the fecal butyrate abundance levels argue that these findings were functionally relevant.

Because beta diversity changes were mild and only significant during the treatment period, we have conveyed the message throughout the manuscript that the microbiota changes were mild and mainly during the treatment period. However, we believe that the findings summarized in the points #1-3 listed above also indicate that there were long-lasting effects that must be reported. In the abstract, the statement that the Reviewer’s suggests to modify is an introductory statement, followed by the description of the specific changes (lines 35-42):

*“Fecal microbiota transplantation (FMT) was safe, not related to severe adverse events, and attenuated HIV-associated dysbiosis. Alpha diversity significantly increased until week 6 in the FMT arm. Beta-diversity changes indicated mild engraftment of donor’s microbiota **during the treatment period**. The greatest engraftment was observed in the participant who had received antibiotics most recently before FMT. The Lachnospiraceae and Ruminococcaceae families were the taxa more robustly engrafted across time-points.”*

Following the previous Reviewer’s suggestion, we have now emphasized that say that beta-diversity changes were transient (“during the treatment period”). Making this consideration in the introductory statement would be misleading, since will make the reader conclude that all changes (including those highlighted above in points #1-3) were transient.

Reviewer #3 (Remarks to the Author):

This reviewer appreciates the detailed response from the corresponding author and study team. Several concerns and questions remain.

- **Can the author confirm that OpenBiome had available butyrate concentrates by donors and that the butyrate level was used to select the 3 donors (line 157)? As**

such, were measurements of butyrate performed in the recipients before and after FMT?

Authors: Yes, we can confirm that Openbiome had available 16S rDNA composition data and fecal butyrate abundance in their donors. The FMT donors were selected because they were in the higher quartile of fecal *Bacteroides* and *Faecalibacterium* genus abundance and butyrate concentrations, and in the lower quartile of *Prevotella* abundance. Indeed, the decision to collaborate with Openbiome was largely based on the possibility of rationally select the microbiota donors based on a microbiota signature.

Unfortunately, the Omnigene Gut kits (DNA Genotek) that were used to collect the fecal samples contain a stabilizer solution that better preserves (relative to RNAlater and Tris-EDTA) the composition of fecal microbial community structure DNA for microbiome analysis (Choo, Leong, & Rogers, 2015), but is not compatible with the liquid-based chromatography assays that are necessary to measure butyrate concentrations. An ongoing analysis from this study will assess the impact of FMT on plasma metabolite levels, and will include the SCFA concentrations in plasma.

• Given the number of tested outcomes (multiple inflammatory markers, markers of microbial translocation, T-cell activation markers, and markers of gut epithelial injury), it seems that any evidence will need to be adjusted by a statistical correction e.g., Bonferroni or a less strict alternative. Alternatively, the result must be interpreted with caution.

Authors: We understand the Reviewer's concern. Please, note that we selected a limited number of biomarkers to be measured in plasma according to their ability to predict mortality, and we explicitly avoided to correlate them with the detected bacterial taxa to avoid the discovery of false correlations between the microbiota and the biomarkers. In contrast, we published the study design in an open repository, specified the outcomes and the statistical methods that would be used (<https://clinicaltrials.gov/ct2/show/NCT03008941> and study protocol approved by the Ethics Committee attached to the submission).

In the methods, we have acknowledged that we did not adjust for multiple comparisons. As it often argued, in pilot studies simply describing what tests have been performed and why is generally the best way of dealing with multiple comparisons (Perneger, BMJ 1999). Throughout the text, we have emphasized that this was a pilot study and that the analysed biomarkers represent exploratory analyses. We have explicitly avoided categorical claims.

To further address this Reviewer concern in the Discussion, we have called to caution when interpreting the results and that the biomarker analysis was not corrected by multiple comparison (lines 560-563):

"We measured 8 biomarkers of inflammation, and found differences between groups only in 1 of them, which could be due to spurious associations, given the small sample sizes assessed and the lack of correction by multiple comparisons".

In the abstract, we have underlined that the IFABP changes were exploratory analyses (line 42):

"In exploratory analyses, significant differences between groups were observed in intestinal fatty acid-binding protein (IFABP)".

• What is the relationship between the patients who experienced the most engraftment and their change in levels of I-FABP?

Authors: The largest IFABP declines occurred in subjects receiving the FMT from Donor A (#2, 4, 29 and 30) and in one subject receiving FMT from Donor B (#23). Please, note the

greatest engraftment (measured as Unifrac distances from donors to recipients) occurred in recipients of Donor A, which reinforces the interpretation that the observed IFABP changes were attributable to the intervention.

- **Would the significance observed in IFABP over time persist if a Wilcoxon signed-rank matched pair test was used? This reduces the effect of outliers e.g., Mann-Whitney.**

Authors: As requested by the Reviewer, we have calculated the differences between the IFABP fold changes from baseline to different timepoints in each group and computed the P values using Wilcoxon rank sum tests.

Table 1. Mean IFABP fold changes from baseline and P values using Wilcoxon rank sum tests.

Changes from baseline	FMT arm	Placebo arm	P Value
To week 1	0.72	0.95	0.0459
To week 2	0.52	0.95	0.0455
To week 8	0.69	0.78	0.803
To week 24	0.67	1	0.295
To week 48	0.77	0.97	0.069

As can be appreciated using this strategy, the findings are consistent with those reported analyses, and even provide lower P values. Because these findings are consistent with those already reported, and the statistical approach originally specified in the approved study protocol for this outcome was the use of linear mixed models, we prefer to keep the more conservative P values obtained from linear mixed models already reported in the manuscript.

- **This reviewer is concerned with the overall conclusion that ‘significant differences between groups were observed in IFABP’, when the results suggest that the difference may have been driven by few individuals (recipients of donor A) with high baseline pre-FMT IFABP levels (figure 6J). If one or two of those individuals were removed from the analysis, would the significance persist? In addition, was the delta IFABP correlated to the delta B-diversity? Consider a more conditional statement if the distribution of change in IFABP is not even across study participants.**

Authors: Given the exploratory nature of the biomarker analyses, we agree with the Reviewer that the differences could be driven by a few individuals. However, we respectfully take issue with the interpretation that outliers in the baseline IFABP values could have biased the results. First, the statistical approach would have mitigated this possibility. We compared trajectories, rather than mean/median values, using linear mixed models which included the interaction term time-versus-treatment group. To allow for correlations caused by repeated observations, we set an autoregressive covariance structure.

Although outliers are often identified subjectively, we have used the Grubbs’ test (extreme studentized deviate method). The baseline IFABP values followed had a mean of 4.7 ng/mL (SD 2.9). The critical Z ratio is 2.9. No observation in this data was reached by this value. Participant #5 had the furthest value from the rest (12.6 ng/mL), but was not a significant outlier (P > 0.05).

To further reassure the Reviewer on this regard, we have re-calculated the P values for the FMT vs. placebo differences from baseline to different timepoints reported in the manuscript (figure 6H) after removal of this subject with the furthest value from the rest. For week 1 the P value increased from 0.063 to 0.106, for week 4 from 0.042 to 0.066 and for week 48 from 0.073 to 0.109. Removal of the second furthest value from the rest (11.3 mg/mL) changed these P values to 0.841, 0.052, and 0.069, respectively. These changes in the P values are not dramatic and are attributable to the loss of statistical power in an analysis with few individuals. Please, consider also the results of the Wilcoxon rank sum tests reported in the answer to the previous comment.

Following the Reviewer's suggestion, we have explored the correlations between changes in beta-diversity (weighted Unifrac distances) and changes in IFABP levels. Because both variables were repeatedly measured and many timepoints were available, to avoid further multiple comparisons we have fitted a linear mixed model with an autoregressive covariance structure in which log₁₀-transformed IFABP concentrations were the outcome and IFABP levels and time the fixed effects, with a random effect for each patient. The P value of the association between IFABP and beta diversity changes was not significant. Please, note that the effects of any FMT intervention could depend on phenomena not captured by changes on beta diversity (such as bacterial metabolic activity), and that in this study we measured directly the effects of the FMT intervention through a randomized placebo-controlled design.

To avoid overstating the significance of the IFABP findings, we have emphasized throughout the text that this was a pilot study, that the analysis of inflammatory biomarkers was exploratory, and we have avoided categorical claims. We have carefully revised the statements regarding the significance of the IFABP changes:

-In the abstract and the results, the statements are essentially descriptive. We have avoided referring to IFABP changes in the statement summarizing the main findings in lines 34-36.

-In the abstract (line 41), we have added before the mention that statistical differences were detected that this was an exploratory outcome.

"In exploratory analyses, significant differences between groups were observed in intestinal fatty acid-binding protein (IFABP)"

-In the opening statements in the discussion that summarizes the main findings of the study, we have avoid the term "significant" from the statement to soften it (line 462).

"While the intervention did not cause any signal of immunological harm, a rapid decline of IFABP was observed in subjects receiving FMT compared to those receiving a placebo".

-In the discussion, section of limitations, we have included a statement to acknowledge that this association between FMT and greater IFABP decline must be interpreted cautiously, given the small sample sizes and we acknowledge that the analysis was not corrected by multiple comparisons (line 635-637).

"We measured 8 biomarkers of inflammation, and found differences between groups only in 1 of them, which could be due to spurious associations, given the small sample sizes assessed and the lack of correction by multiple comparisons".

REFERENCES

Choo, J. M., Leong, L. E. X., & Rogers, G. B. (2015). Sample storage conditions significantly

influence faecal microbiome profiles. *Scientific Reports*, 5, 16350.
<https://doi.org/10.1038/srep16350>

Vujkovic-Cvijin, I., & Somsouk, M. (2019). HIV and the Gut Microbiota: Composition, Consequences, and Avenues for Amelioration. *Current HIV/AIDS Reports*.
<https://doi.org/10.1007/s11904-019-00441-w>

RESPONSE TO REVIEWERS

Manuscript number: NCOMMS-20-21964A

Article title: A pilot placebo-controlled study of fecal microbiota transplantation to target inflammation in HIV: A pilot, randomized, placebo-controlled study

The authors would like to thank the Reviewers and Editors for their careful review of our manuscript, and for providing us with their very helpful comments and suggestions to improve the quality of the manuscript.

REVIEWERS' COMMENTS

Reviewer #3 (Remarks to the Author):

Concerns have been addressed